# Surveillance of $SO_2$ and $NO_2$ from ship emissions by MAX-DOAS measurements and implication to compliance of fuel sulfur content

**Yuli Cheng[1], Shanshan Wang[1,2,*], Jian Zhu[1], Yanlin Guo[1], Ruifeng Zhang[1], Yiming Liu[1], Yan Zhang[1,2,3], Qi Yu[1,2], Weichun Ma[1,2] and Bin Zhou[1,2,3,*]**

[1] Shanghai Key Laboratory of Atmospheric Particle Pollution and Prevention (LAP[3]), Department of Environmental Science and Engineering, Fudan University, Shanghai, China

[2] Shanghai Institute of Eco-Chongming (SIEC), No.3663 Northern Zhongshan Road, Shanghai, 200062, China

[3] Institute of Atmospheric Sciences, Fudan University, Shanghai, 200433, China

*Correspondence to*: Shanshan Wang (shanshanwang@fudan.edu.cn) and Bin Zhou (binzhou@fudan.edu.cn)

**Abstract.**

With the increased concerns on the shipping emitted air pollutants, the feasible technology for the surveillance is in high demand. Here we presented the shore-based Multi-Axis Differential Optical Absorption Spectroscopy (MAX-DOAS) measurements of emitted $SO_2$ and $NO_2$ from ships under different traffic conditions in China's ship emission control area (ECA) of Shanghai and Shenzhen, China. Three typical measurement sites were selected in these two regions to represent emission scenarios of ships docked at berth, navigation in the inland waterway and inbound/outbound in the deep water port. With 2-dimensional scanning, the observation shows that the hotspots of $SO_2$ and $NO_2$ can be quickly and easily located from multiple berths. Although the MAX-DOAS measurements can not distinguish the single ship plume in the busy shipping lanes of inland waterway area, it certificates that the variations of $SO_2$ and $NO_2$ levels are mainly impacted by the ship traffic density and atmospheric dispersion conditions. In the open water area with low density of vessels, the MAX-DOAS measurements can capture the pulse signal of ship emitted $SO_2$ and $NO_2$ very well, and characterize the peaks altitude and insistent duration of the individual ship plumes. Combined with the ship information of activity data, rated power of engine and fuel sulfur content, it was found that the $SO_2/NO_2$ ratio in single plume is usually low (<1.5) for inbound vessel due to the usage of auxiliary engine with less power and clean fuel of low sulfur content. Meanwhile, the unexpected high $SO_2/NO_2$ ratio implies the fuel usage with sulfur content exceeding limit of regulations. Therefore, the observed $SO_2/NO_2$ ratio in the plume of single ship can be used as the index for the compliance of fuel sulfur content, and then tag the suspicious ship for further enforcement. Combining the ship emission estimated by actual operation parameters and logical sulfur content, the shore-based MAX-DOAS measurement will provide the fast and more accurate way for the surveillance of ship emissions.

## 1 Introduction

Sulphur dioxide ($SO_2$) and nitrogen dioxide ($NO_2$) are the important air pollutants, and also recognized as the non-negligible pollutants of ship emissions (Corbett et al., 1999; Endresen et al., 2003; Eyring et al., 2010; Matthias et al., 2010). Both of

them can engage in the atmospheric chemical reactions to produce aerosols and acid rain, and further have negative effects on the air quality, climate system, and human health, as well as acidification of terrestrial and marine ecosystem (Berglen et al., 2004; Seinfeld and Pandis, 2006; Singh, 1987). Moreover, $NO_2$ is also the key substance to form photochemical smog (Dimitriades, 1972). With the rapid growth of the transportation volume, air pollution has become the most challenging environmental issue in the shipping industry, such as the emissions of $CO_2$, $SO_2$, NOx, particles and greenhouse gases. $CO_2$ and NO are the main pollutants emitted by ships, and NO is rapidly converted to $NO_2$ by reaction with $O_3$. (Eyring et al., 2005; Becagli et al., 2012; Coggon et al., 2012; Diesch et al., 2013; Lauer et al., 2007; Seyler et al., 2017). Eyring et al. (2010) reported that ships contribute 15% of global NOx emissions and 4-9% of $SO_2$, respectively. In the view of spatial distribution, global hotspots with high $SO_2$ and $NO_2$ emissions were identified to be the regions in Eastern and Southern China Seas, the sea areas in the south-eastern and southern Asia, the Red Sea, the Mediterranean, North Atlantic near the European coast, and along the western coast of North America, etc. (Johansson et al., 2017). In China, ship emitted pollutants play important roles in the air quality, human health and climate (Lai et al., 2013; Liu et al., 2016; Yang et al., 2007). It not only affects the air quality in coastal areas, but even influences the inland areas hundreds of kilometers away from the emission sources (Lv et al., 2018). The port city is most affected by ship pollution, followed by cities along the river. As an example, ship emitted $SO_2$ and NOx occupied for 12.4% and 11.6% of total emissions of the whole city of Shanghai in 2012, respectively, while there could be 64% of primary $PM_{2.5}$ contributed by ships in Shanghai Port transported to inland region (Fan et al., 2016; Zhao et al., 2013).

In order to reduce the negative impacts of ship emissions, the European Union and the United States have implemented regulations in an effort to decrease ship emissions (Kattner et al., 2015), among which the fuel quality regulation has been proven to be the most effective measures for addressing the issue of sulphur oxides (SOx) and particulate matter (PM) emissions in many countries. Besides, the International Maritime Organization (IMO, 2009) also has set up multiple emission control areas (ECA) worldwide. By 2020, the maximum FSC is 0.50% S m/m all over the world, and it is worth noting that the maximum fuel sulfur content in ECAs at the US coast and Europe is 0.10 % S m/m, while in China's ECA it is 0.50 % S m/m. The regulations also set limits for pollutant emissions such as NOx and $CO_2$ in the exhaust gas. Alternatively, the exhaust gas treatment system could be another option. Since January 1st, 2017, ships berth at the core ports of three designated Domestic Emission Control Area (DECA) in Pearl River Delta (PRD), the Yangtze River Delta (YRD) and the Bohai Rim (Beijing-Tianjin-Hebei area) of China should use fuel with sulphur content less than or equal to 0.50% (MOT, 2015). As of January 1, 2019, the ship entering the ECAs should use fuel with a sulfur content of not more than 0.50 % m/m, whether it is sailing or docking. Currently, the scrubber is also the alternative way to reduce the ship emission in China. As a consequence, it is obviously that the reliable and practical monitoring system are highly demanded for the implementation of ECA regulations.

The common options to monitor ship emissions can be classified into two categories: estimates based on activity data or written documentation; and measurements of on-board fuel sample and exhaust gas made on board the ship. Basically, the continuous online monitoring of fuel and exhaust gas on-board is the highly effective and accurate supervision means, but less operability in practice. For the regulatory party, fuel sampling and document inspection are currently the common measures and the sulfur

content in the fuel is usually fast detecting after the ship is docked in dozen of minutes. Besides, other technical methods has been developed to determine both $SO_2$ and NOx emissions, such as a new type of ship exhaust gas detection technology that mounts a portable sniffer/instrument on board a ship or on a helicopter (Beecken et al., 2015; Berg et al., 2012; Murphy et al., 2009; Villa et al., 2016). Alternatively, shore-based remote sensing is another effective way to measure the ship plume and further estimate the sulfur content, when ships pass the lanes or dock at berth (Kattner et al. 2015; Seyler et al. 2017).

Remote sensing technique shows the advantages of fast detection, easy operation and high automation. Besides the passive "sniffing" method with in-situ instrumentation, optical remote sensing technique can detect the variation of the light properties after interaction with the exhaust plume and corresponding $SO_2$ and $NO_2$ emission in the plume, such as differential optical absorption spectroscopy (DOAS), light detection and ranging (LIDAR), and the ultraviolet camera (UV-CAM) technique

(Balzani et al., 2014; Seyler et al., 2017). LIDAR system can be used to retrieve a 2-dimentional concentration distribution by scanning through the ship plume, and to obtain the ship emissions combing the wind and concentration profiles. McLaren et al. (2012) employed active long-path DOAS technique to measure $NO_2$-to-$SO_2$ ratios in ship plumes and speculate on its relationship with the sulfur content of fuels. UV camera has been successfully applied to measure the $SO_2$ concentrations and emission rates of moving and stationary ship plumes (Prata, 2014).

DOAS technique allows to identify and quantify the absorption of variety of species showing characteristic absorption features in the wavelength range (Platt and Stutz, 2008). It has been widely used for trace gases measurements in several decades, especially very mature for $NO_2$ and $SO_2$ (Edner et al., 1993; Mellqvist and Rosén, 1996; Platt et al., 1979). As an expanded apparatus, the multi-axis differential optical absorption spectroscopy (MAX-DOAS) measurements are high sensitivity to

aerosols and trace gases in the lower troposphere by observing scattered sunlight under different viewing angles close to the horizontal and the zenith directions (Hönninger et al., 2004; Ma et al., 2013; Sinreich et al., 2005; Wang et al., 2014). Due to its portability, MAX-DOAS instrument can be carried on the ship to observe the vertical column densities (VCDs) of $NO_2$ and $SO_2$ along the cruise, during which high levels of pollutants were found close to the busy port and dense lanes (Hong et al., 2018; Schreier et al., 2015; Takashima et al., 2012; Tan et al., 2018). In addition, MAX-DOAS has been successfully employed

for monitoring shipping emissions directly. Premuda et al. (2011) used the ground-based MAX-DOAS measurements to evaluate the $NO_2$ and $SO_2$ levels in ship plume discharged from the single ship in the Giudecca Strait of the Venetian Lagoon. Seyler et al. (2017) have utilized MAX-DOAS to perform long-term measurements of $NO_2$ and $SO_2$ from shipping emissions in the German Bight, and evaluated the reduction in $SO_2$ levels after implement stricter sulfur limits in shipping fuel.

In this study, the shore-based MAX-DOAS measurements were employed to measure the $NO_2$ and $SO_2$ in ship plumes for different ship traffic environments of Shanghai and Shenzhen, China. Combined with the photos taken by the camera of the instrument and AIS (Automatic Identification System) information, it is verified that emissions of ships at berth, passing through the inland waterway and open sea areas can be successfully detected. The measurements can also provide the fuel sulfur content information of individual ship by comparing the emitted $NO_2$ and $SO_2$ in the plume considering the effects of plume age. With the fuel sample analysis and ship activity data, it suggests that the shore-based MAX-DOAS method shows the feasibility and reliability of surveillance for ship emitted $SO_2$ and $NO_2$ and further allows to know compliance of fuel sulfur content.

## 2 Measurements and method

### 2.1 Instrument setup and sites

In this study, the MAX-DOAS instrument has been designed and assemble by the authors (Zhang et al., 2018a). It mainly consists of a receiving telescope, a spectrometer (Ocean Optics, QE65 Pro), and a computer to control the measurements. Driven by the two-dimensional stepper motor system, the telescope can collect the scattered sunlight from different elevation angles in vertical and azimuth angles in horizontal. The scattered light is converged by the lens to the fiber bundle connected to the spectrometer. The receiving sunlight is dispersed by a grating, detected by a CCD detector and recorded by the spectrometer covering the wavelength range from 300 to 480 nm with a resolution about 0.5 nm Full Width at Half Maximum (FWHM). As a new designed feature, a camera has been configured on the MAX-DOAS apparatus, which moves coaxially with the receiving telescope and can record the scene and sky conditions same as the views of telescope. The scanning of telescope can be set in the sequence of several elevation angles from close to horizontal and 90° and then move to next azimuth angle for another vertical scanning sequence. Due to the different ship traffic conditions, the types of ship passing in inland waterway and seaside ports are different in size and tonnage. Therefore, the configuration of observing geometric angels were adjusted dependent on the conditions of ships, as referred in Table 1.

The MAX-DOAS measurements of ship emissions were performed in two typical port cities of Shanghai and Shenzhen in China. As shown in Fig. 1(a), sea areas in surrounding Shanghai and Shenzhen city are located in the ECA of Yangtze River Delta and Pearl River Delta. In Shanghai, two different ship traffic scenarios were considered, i.e. ships at berth in Waigaoqiao container port area (31.36° N, 121.58° E, Fig. 1(b)), and ships passing through inland waterway of the downstream of Huangpu River at Wusong area (31.37° N, 121.50° E, Fig. 1(c)). In Shenzhen, the measurements were carried out in the deep water port of Yantian (114.29° E, 22.56° N, Fig. 1(d)). More details about the environments and operation configurations of measurement were listed in Table 1.

**Table 1. Measurements details and operation configurations of MAX-DOAS.**

| Sites | Locations and periods | Operations* | Environment types |
|---|---|---|---|
| Waigaoqiao, Shanghai | 31.36° N, 121.58° E 28/08/2017 | AZ: 26° to 34°; ELE: 3°, 4°, 5°, 6°, 7°; Spectrum temporal resolution: 15-30 s; Completed scanning cycle: 15 min. | Viewing to: berths; Ships: container ship. |
| Wusong, Shanghai | 31.37° N, 121.50° E 30/12/2017-18/05/2018 | AZ: 85°; ELE: 0°, 1°, 2°, 3°, 4°, 5°, 6°, 8°, 65°; Spectrum temporal resolution: 40 s; Completed scanning cycle: 7 min. | Viewing to: inland waterway with high traffic volume; Ships: a wide variety of ships and small in size |
| Yantian, Shenzhen | 114.29° E, 22.56° N 23/05/2018-30/06/2018 | AZ: 75°; ELE: 2°, 3°, 5°, 7°, 10°, 15°, 30°, 90°; Spectrum temporal resolution: 60 s; Completed scanning cycle: 9 min. | Viewing to: open sea areas with smaller traffic volume; Ships: container ship as the main part |

*AZ: azimuth angle in horizontal direction; ELE: elevation angle in vertical direction.

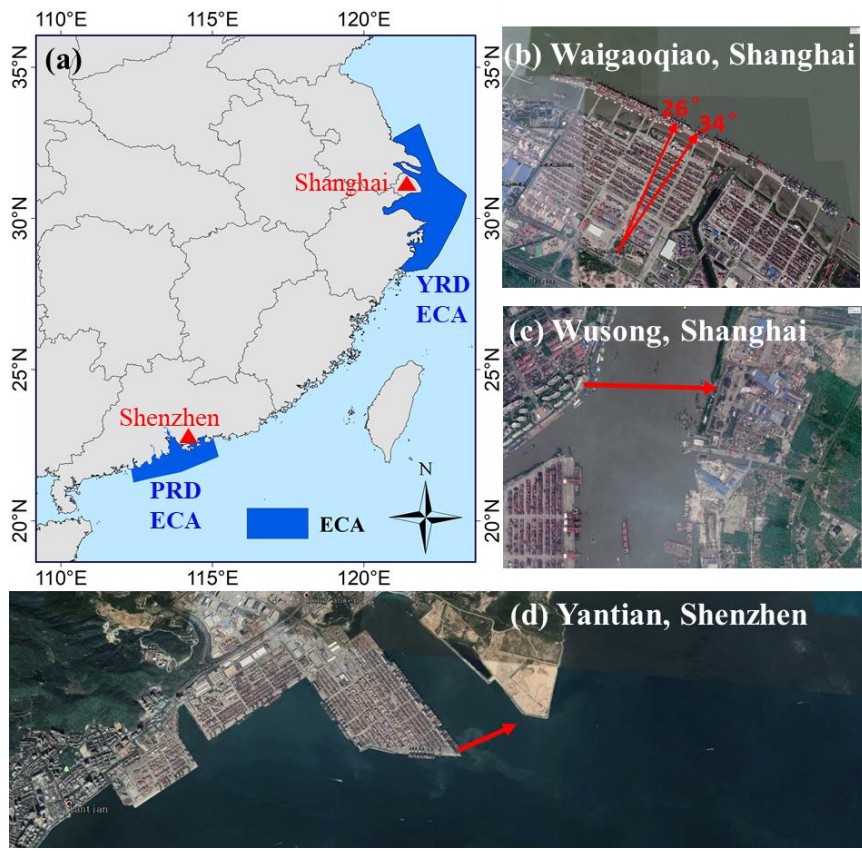


**Figure 1. The YRD and PRD Domestic Emission Control Area (DECA) in China and locations of the MAX-DOAS measurements in coastal cities of Shanghai and Shenzhen: Waigaoqiao Port and Wusong Wharf in Shanghai, and Yantian Port in Shenzhen. The viewing direction of instrument azimuth angle is indicated by a red arrow. (b) to (d) cited from Google Earth.**

**2.2 DOAS spectral analysis**

The algorithm of DOAS is basically based on Lambert-Beer law, which describes the extinction of radiation through the atmosphere (Platt and Stutz, 2008). MAX-DOAS instrument observes scattered sunlight from various viewing directions and records the spectrum (e.g. Hönninger et al., 2004; Sinreich et al., 2005).The spectral analysis generates the measured SCD (slant column densities), defined as the integral of the trace gas concentration along the entire optical path including the SCDs

in the troposphere and the stratosphere. (e.g. Platt and Stutz, 2008, Wagner, et al., 2010). The stratospheric absorption has been assumed as the same level in all spectra taken within one scan cycle, so we generally choose the spectrum with less trace gas absorption as the reference, such as the spectrum measured in zenith direction. The slant column concentration of the trace gas measured at each lower elevation angle ($\alpha$) is represented by DSCD (differential SCD), which is the gas information of the measured slant column densities minus background densities in the reference spectrum.

DSCD (α) = SCD (α) – SCD (ref)

$$= SCD (\alpha)_{trop} + SCD_{strat} - SCD (ref)_{trop} - SCD_{strat}$$

$$= SCD (\alpha)_{trop} - SCD (ref)_{trop} \qquad (1)$$

Based on the DOAS principle, the measured scattered sun-light spectra are analyzed using the QDOAS spectral fitting software, which is developed by BIRA-IASB (http://uv-vis.aeronomie.be/software/QDOAS/). The strong absorption band of $SO_2$ is

below 325 nm, where the $NO_2$ absorption are relatively weak. Therefore, the fitting wavelength intervals of $SO_2$ and $NO_2$ are 307.5-320 nm and 338-370 nm, respectively. Although the generated DSCDs are related to the wavelength of fitting interval, the usage of ratio of $SO_2$ to $NO_2$ DSCDs retrieved at different spectral ranges will not impacted by the effect of wavelength dependency. Trace gases with absorptions in relevant fitting windows and Ring spectrum were included. The details of spectral fitting configuration are listed in Table 2. Wavelength calibration was performed by using high-resolution solar reference

spectrum (Chance and Kurucz, 2010). The offset and the signal of the dark current were measured every night and extracted automatically from the measured spectra before spectral analysis. Consequently, the DSCDs of $SO_2$ and $NO_2$ were yielded by taken the measured spectrum at 90° as the Fraunhofer reference spectrum.

**Table 2. Configuration of spectral fitting for $SO_2$ and $NO_2$**

| Parameters | $SO_2$ | $NO_2$ |
|---|---|---|
| Fitting window | 307.5–320 nm | 338–370 nm |
| $NO_2$ | 298 K (Vandaele et al., 1998) | |
| $SO_2$ | 293 K (Bogumil et al., 2003) | / |
| $O_4$ | / | 293 K (Thalman and Volkamer, 2013) |
| $O_3$ | 223 K & 243 K (Serdyuchenko et al., 2014) | 223 K (Serdyuchenko et al., 2014) |
| BrO | / | 293 K (Fleischmann et al., 2004) |
| $CH_2O$ | 298K (Meller and Moortgat, 2000) | |
| Ring | Calculated by QDOAS | |
| Polynomial Degree | 3 | 5 |
| Intensity Offset | Constant | |


Figure 2 shows the typical spectral fitting of measured spectra with and without ship plume contamination. The obvious absorbing structures of $SO_2$, $NO_2$ and fairly low residuals can be observed in both conditions of polluted spectrum (collected at elevation angle of 5° at 10:39 LT on 22 June, 2018) and clean spectrum (collected at elevation angle of 5° at 09:53 LT on 22 June, 2018). By contrast, the retrieved $SO_2$ and $NO_2$ DSCDs of $8.11 \times 10^{16}$ and $3.08 \times 10^{16}$ molec $cm^{-2}$ in polluted case are

significantly higher than those in clean condition of $2.24 \times 10^{16}$ and $1.61 \times 10^{16}$ molec $cm^{-2}$. It demonstrates the high sensitivity of measurements to ship plumes and the good performance of the spectral fitting. In this study, a threshold of residual $< 1 \times 10^{-}$

[3] is used for screen out the unsatisfied spectral fitting, and 94.57% of $NO_2$ DSCDs results and 76.26% of $SO_2$ fitting results remains in further discussion. The uncertainties for the spectral analysis of $SO_2$ were higher because of the weak scatter sunlight intensity and lower signal-to-noise at the short wavelengths.

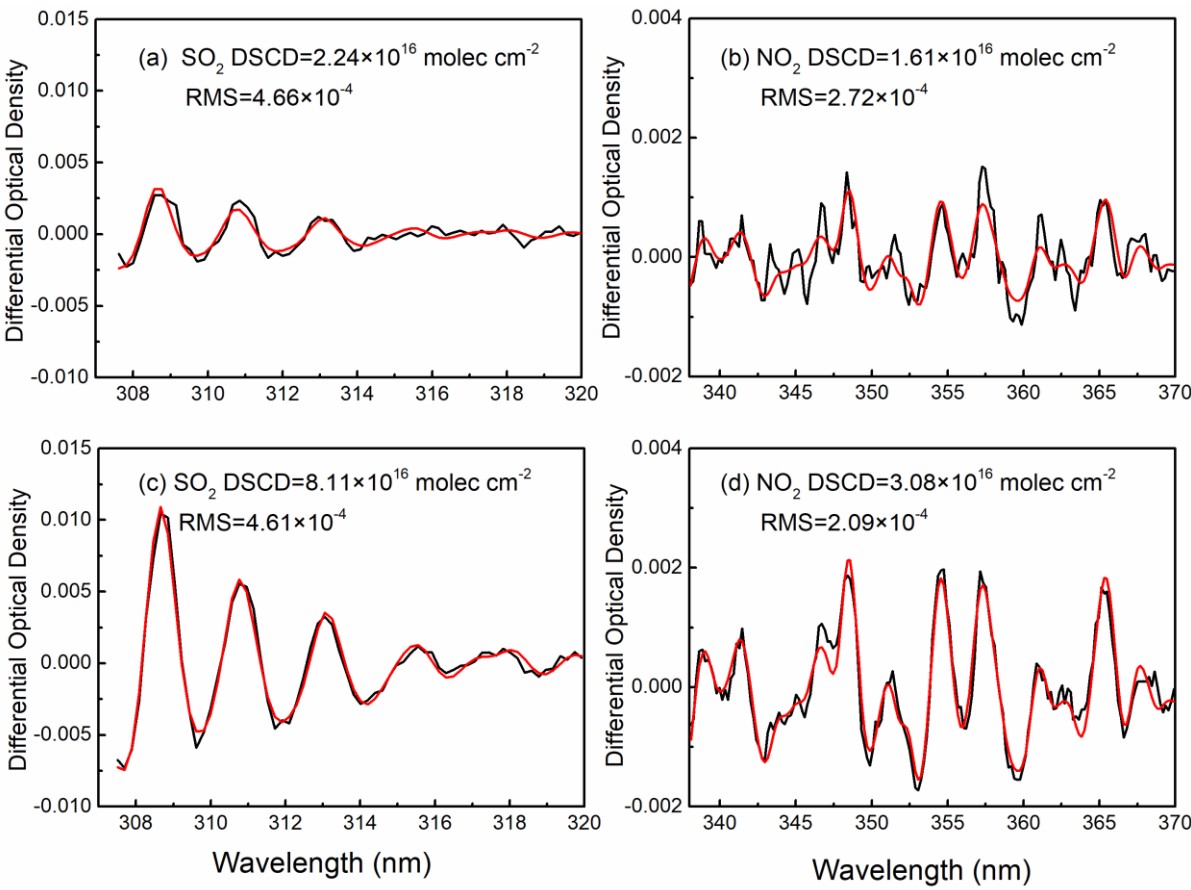


**Figure 2. Typical DOAS spectral fitting for SO₂ and NO₂. (a) and (b) show the clean condition of spectrum collected at an elevation angle of 5° at 10:39 LT on 22 June, 2018, while (c) and (d) are the ship plumes polluted case of spectrum measured at an elevation angle of 5° at 09:53 LT on 22 June, 2018. Black lines show the measured atmospheric spectrum and the red line shows the reference absorption cross-section.**

**2.3 MAX-DOAS measurements for ship emissions**

For ship emissions measurements, the DSCDs of pollutants at low elevation angles should express the change of integrated concentrations along the light path after contamination by the exhaust plume, which collects scattered sunlight passing through ship plumes. Figure 3 (a) depicts the schematic diagram of ground-based MAX-DOAS measurements of ship emissions. The telescope is pointed towards the ship lanes or the direction of ship passed through. Consequently, the measured spectra at low

elevation angle will be impacted by the plumes of ship emissions.

In order to better demonstrate the background concentration of $NO_2$ and $SO_2$, several typical cycles on June 29 were selected as examples. Figure 3 (b) and (c) show the vertical distributions of $NO_2$ and $SO_2$ DSCDs with the elevation angle when there is a ship passing through and not. It can be observed that the DSCDs of $NO_2$ and $SO_2$ decrease slowly with increasing angle under clean conditions, during which the maximum values of $NO_2$ and $SO_2$ DSCDs are $5.03\times10^{16}$ molec $cm^{-2}$ at elevation $3°$ and $1.78\times10^{16}$ molec $cm^{-2}$ at elevation $2°$, respectively. In contrast, the $NO_2$ and $SO_2$ DSCDs increased significantly when ships passed, showing the maximum values of $NO_2$ and $SO_2$ DSCDs of $7.36\times10^{16}$ molec $cm^{-2}$ at elevation $5°$ and $4.15\times10^{16}$ molec $cm^{-2}$ at elevation $5°$, respectively. And the highest value of $SO_2$ generally appears between elevation angle $5°$ and $10°$.

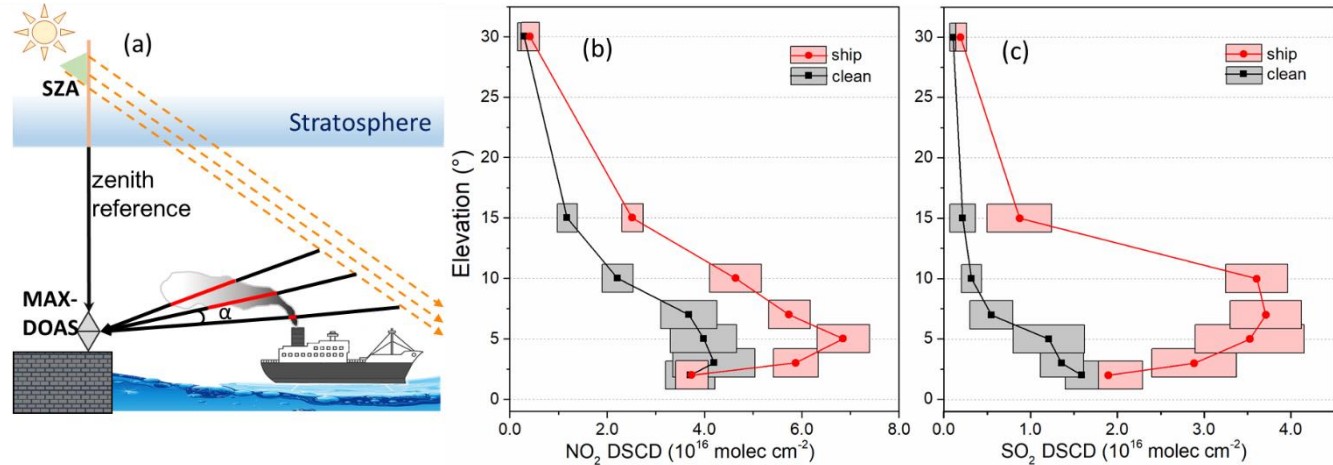

**Figure 3. (a) Schematic diagram of MAX-DOAS measurement geometry for monitoring ship emissions, and the distributions of (b) $NO_2$ and (c) $SO_2$ DSCDs with elevation angle on June 29, 2018.**

## 3 Results and discussion

### 3.1 Identifying the emissions of ship at berth

The measurement site at Waigaoqiao Container Terminal is located on the south bank of the Yangtze River, and close to the confluence of Yangtze River and Huangpu River. The Terminal has a total quay length of more than one kilometer, and its three container berths are able to accommodate the fifth and sixth generation container ships. The special location determines that it is the important traffic route for ships to enter or leave the Yangtze River, Shanghai Port, and Waigaoqiao wharf. There are no other obvious sources for $SO_2$ and $NO_2$ than emissions from ships. In order to detect the emissions of ship at berths, the MAX-DOAS instrument is placed on a fifth floor at the building of Pudong MSB (Maritime Safety Bureau). The distance between the instrument and berth is about 1.4 km. Since no other constructions obscured, multiple berths can be seen directly in the viewing of the MAX-DOAS instrument. Considering the size and chimney height of berthed ships, the MAX-DOAS

telescope was set to scan vertically in sequence of 3°, 4°, 5°, 6°, and 7° (indicated with angle α in Fig. 4). In horizontal, telescope ranged from 26° to 34° (the viewing angle from north in clockwise) and yielded a range of angle β, which covers about 195 m quay length. After completing one full scanning in both vertical and horizontal, a 2-D distributions of DSCDs in front view of the instrument can be generated. To avoid the interference of pollutants absorptions in the reference spectrum, the spectrum measured at azimuth angle of 10° was considered as the reference spectrum in the background area without the direct ship emissions pollution.

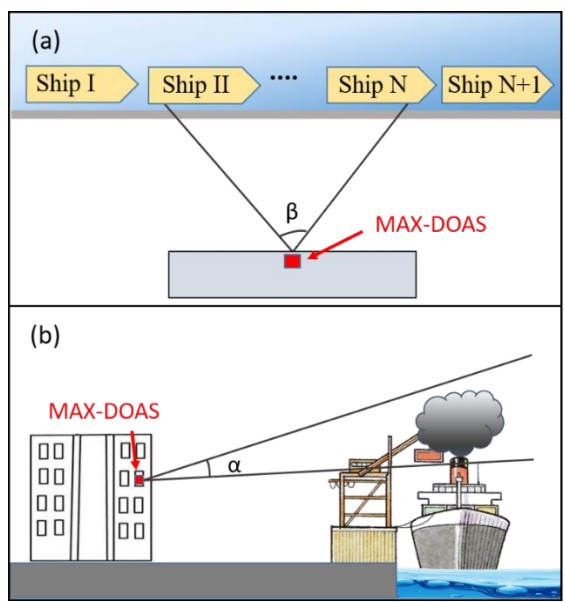

Figure 4. The observational geometry of MAX-DOAS for identifying the emissions of ship at berth in Waigaoqiao port, Shanghai: (a) top view and (b) side view.

Figure 5(a) and (b) shows the spatial distributions of $SO_2$ and $NO_2$ DSCDs in horizontal and vertical during a complete scanning sequence, respectively. Large spatial gradients can be observed for both $SO_2$ and $NO_2$ levels, while the pollutants concentrations are in general higher at lower elevation angels and declined with the increase of height. The highest $SO_2$, i.e. DSCDs up to $2.5 \times 10^{16}$ molec cm$^{-2}$, appeared in horizontal azimuth of 31° and elevation of 3° and attenuated in the direction toward left-upward. Similarly, hot-spots of $NO_2$ with DSCDs of $7.0 \sim 8.0 \times 10^{16}$ molec cm$^{-2}$ are centered between 31° and 33° in horizontal at elevation of 3°, and decreases in periphery. It should be noted that the hot-spots of $SO_2$ and $NO_2$ distribution are shifted to the left accordingly while the height is raised. It is implied that the plumes containing $SO_2$ and $NO_2$ emitted at the bottom in the observational field of view, dispersed and diluted in left-up ward, and the weather recorded that the wind at this test site mainly came from the south. Combined with the real scene shown in Fig. 5(c), the rectangle encircled by dash line indicates the range of MAX-DOAS telescope scanning. It can be seen that there are smoke clusters discharged by ship at the right part of the picture, which is correspond to the azimuth angle between 31° and 33°. And under the action of the wind, the plume spreads to the left of the observational view.

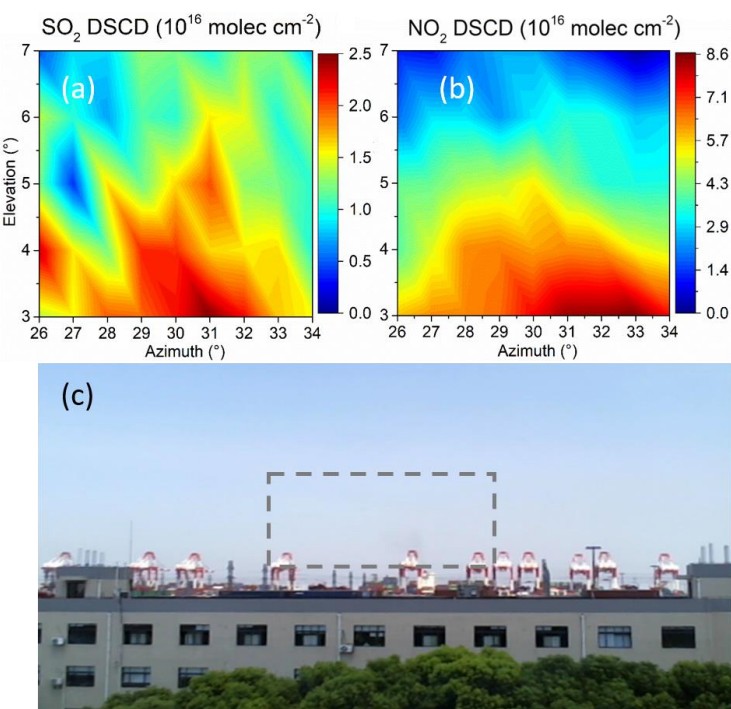


**Figure 5. Distributions of measured DSCDs of (a) SO₂ and (b) NO₂ from emissions of ship at berth during 12:04~12:20 in 28 August, 2017 and (c) live photo captured by camera of MAX-DOAS instrument. The dash line rectangle indicates the observational view of MAX-DOAS.**

Since a full 2-D scanning sequence in horizontal and vertical directions took about 15 minutes, more than a dozen cycles in total can be performed during the afternoon. In view of the identified emission source position above, considering that the hotspots of NO₂ and SO₂ in each 2-D distributions are related to the azimuth of the berth where the ship is docked and the corresponding ship operation status, the DSCDs of NO₂ and SO₂ observed at elevation 4° and azimuth angle between 31°-33° were selected to display the temporal pattern of emissions at berth without averaging. In general, the level of NO₂ DSCDs is
much higher than SO₂, because there are considerable NOx emission of in port trucks between the berth and the instrument, whereas there no other obvious emission source of SO₂. In order to show the variations of DSCDs with less interference due to light path change, we used the mathematic method to remove the slowly change from the trend line of NO₂ and SO₂ DSCDs, and kept the residual after background subtraction in the Fig. 6. The baseline is modeled as a low-pass signal, while the series of peaks is modeled as sparse with sparse derivatives. Moreover, to account for the positivity of peaks, both asymmetric and
symmetric penalty functions are utilized (Ning et al., 2014). Afterwards, four significant increases of SO₂ levels can be observed during this afternoon, accompanied with NO₂ enhancements at approximately same moment, which can be verified by the real scene photos showing evidently the emitted plumes from the expected exhausting position.

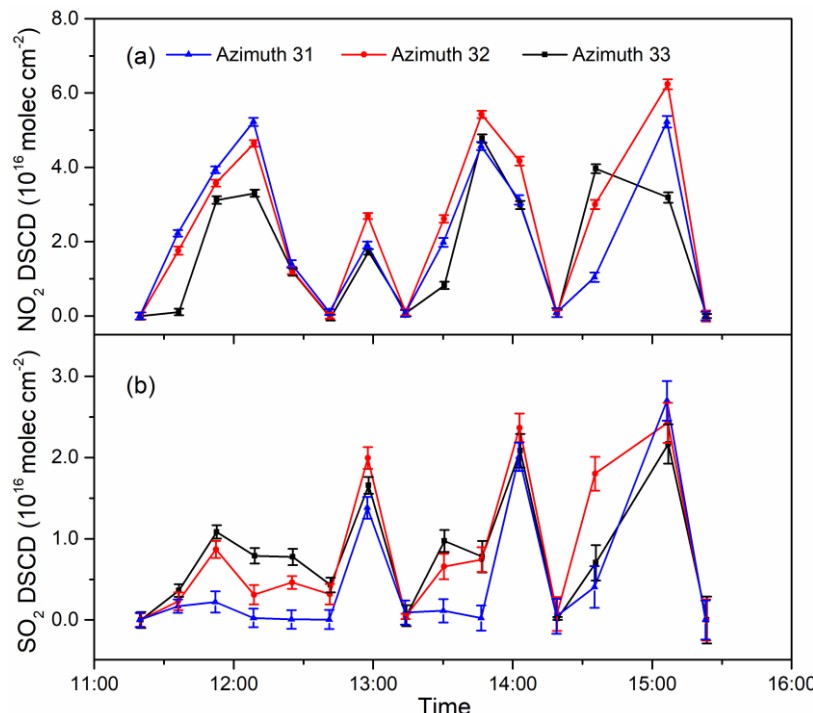

**Figure 6.** Time series of DSCD of (a) $NO_2$ and (b) $SO_2$ measured at 4° elevation angle in three azimuths on August 28, 2017.

Therefore, it can be concluded that the concentration of $NO_2$ and $SO_2$ gases contained in the plume emitted by the container ships and the corresponding discharge position at berth were identified and monitored remotely by the 2-D MAX-DOAS observation. This application of 2-D MAX-DOAS is similar to the Imagine DOAS (IDOAS) technique, which is also used to map the 2-dimensional spatial distribution of polluted gases, such as the distribution of $SO_2$ in plumes of the industrial point sources (General et al., 2014; Pikelnaya et al., 2013). It suggests that the 2-D DOAS technique has the potential to measure the polluted gases mapping from the ships.

### 3.2 Ship emissions at inland waterway

Besides of ocean-going ships, inland waterway vessels also contribute significantly to the amount ship emissions (Kurtenbach et al., 2016; Pillot et al., 2016; Zhang et al., 2017). To consider this situation, the MAX-DOAS instrument was installed at the outside windowsill on the third floor of Wusong MSB Building (121°29′ E, 31°22′ N) from December 30, 2017 to May 18, 2018. The measurement site is located in the downstream of Huangpu River and close to the confluence into the Yangtze River. It is only channel to the upstream of Huangpu River. There are some non-container terminals near the measurement site, which mainly handles goods in domestic trading. As a consequence, a large number of ships entering and leaving the wharf area every day, and the lanes in the downstream of Huangpu River suffers from dense ship traffic. By checking the synchronized

photos taken by the camera attached to the instrument, it is found that the types of vessels passing through are in a wide variety, such as medium and small sized container ships, passenger vessels, bulker and cargo ships. In addition, the traffic volume in the area is quite high, even up to hundreds of ships per hour. As shown in Fig. 1(c), the view direction of MAX-DOAS measurements in Wusong area is pointed perpendicularly to the river lane. The observed signal of pollutants mainly come

from the emissions of ships in navigation. Nevertheless, a small dock and a station for transport container are located opposite the river, which might slightly influence the measurements. The elevations angles were set in scanning sequence of 0°, 1°, 2°, 3°, 4°, 5°, 6°, and 8°. The spectrum measured at 65° was utilized as the reference since the zenith direction is blocked to some extent by the MSB building.

In order to illustrate the impact of ship traffic volume and meteorological conditions, measurement data of 30-min averaged wind speed, observed $NO_2$ and $SO_2$ DSCD on two representative days of January 1, and March 9, 2018 were shown in Fig. 7, as well as the corresponding number of passing ships. It can be seen from Fig. 7(a) that January 1 and March 9 were selected to represent days under stabile and unstable atmospheric conditions. The average $SO_2$ DSCDs changed from $1.8 \times 10^{16}$ to $3.0 \times 10^{16}$ molec cm$^{-2}$, while average $NO_2$ DSCDs varied between $4.0 \times 10^{16}$ and $1.1 \times 10^{17}$ molec cm$^{-2}$ on January 1. On March

9, the $SO_2$ and $NO_2$ average DSCDs ranged $1.0 \times 10^{16}$ to $2.7 \times 10^{16}$ molec cm$^{-2}$ and $2.5 \times 10^{16}$ to $1.0 \times 10^{17}$ molec cm$^{-2}$, respectively. According to the ship traffic density shown in Fig. 7(b), the diurnal variations of $SO_2$ and $NO_2$ DSCDs are obviously closely related to the flow of the ships (quantitative information). Although the averaged $SO_2$ and $NO_2$ DSCDs levels are comparable during these two days, the ship traffic flow on March 9 was overall 50% higher than January 1, which may imply the important role of meteorological conditions. Considering the dense ship lanes, the ship emitted pollutants are easily to be accumulated

under the unfavorable condition with lower wind speed less than 2 m·s$^{-1}$ on January 1. In the contrary, the ship emissions along the lanes can be spread for better diffusion when the averaged wind speed is around 5 m·s$^{-1}$ on March 9.

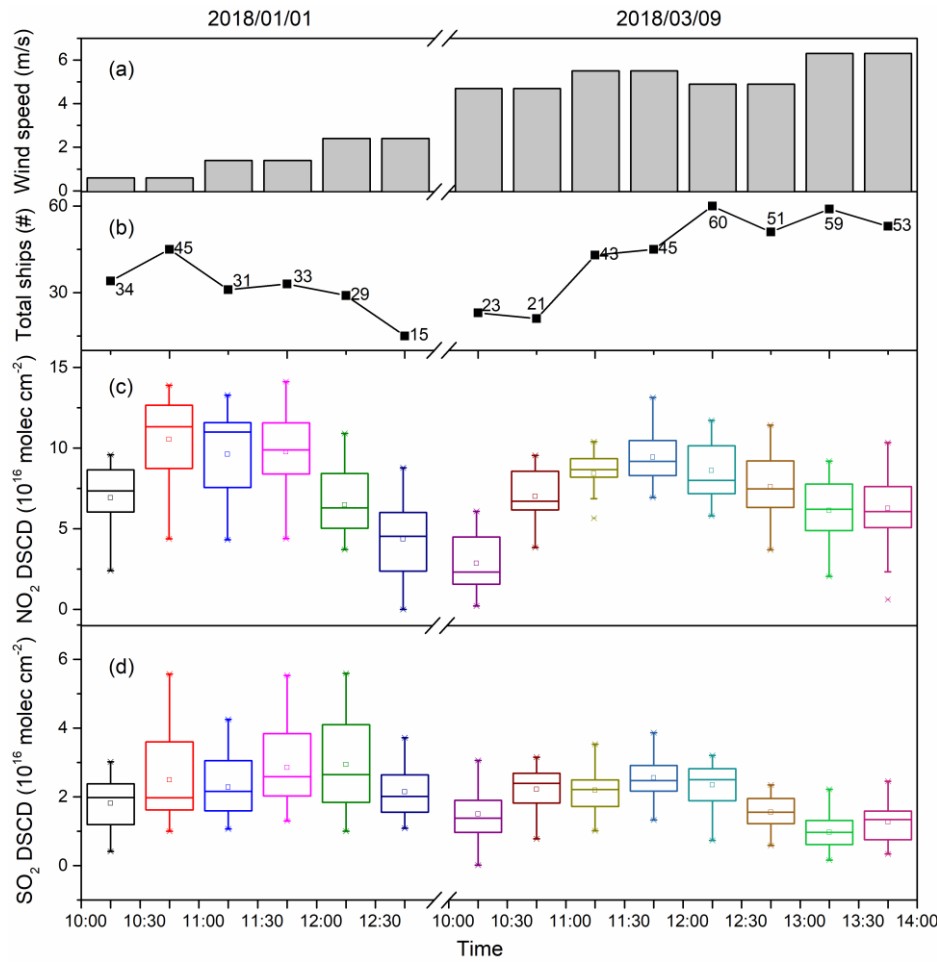

**Figure 7. 30-min averaged wind speed, ship traffic volume, observed NO₂ and SO₂ DSCDs from 10:00 to 13:00 on January 1 and 10:00 to 14:00 on March 9 at Wusong Wharf measurement site.** The hollow squares in the middle of the box represent the mean value, and the solid lines in the middle represent the median. Whiskers extend from each end of the box to the internal and external limits. "-" represents the maximum and minimum, and "×" are 1% and 99% quantiles. The upper and lower edges of the box are 25% and 75% quantiles, respectively.

In order to investigate the impacts of wind on the observed DSCDs, the wind rose diagrams of NO₂ and SO₂ DSCDs at elevation 5° from January to March 2018 are shown in of Fig. 8(a) and (b). It can be found that the wind mainly comes from NNW during the observation period. The average of NO₂ DSCDs is small under the wind conditions from North. When the wind direction is parallel to the observation direction (i.e. E and W, the viewing direction of the telescope is pointing to the East), the average DSCDs of NO₂ is significantly higher. Similarly, the averaged SO₂ DSCDs under the east and west wind is higher than that in the S and N. It suggests that the optical length inside the polluted air and therefore the response signal is probably increased when the wind transports the polluted air parallel to the DOAS viewing direction. In Figure 8 (c) and (d), the

perpendicular direction for N and S is considered wind from 0°±15° and 180°±15°, and the parallel direction for E and W is considered wind from 90°±15° and 270°±15°. It can be seen that the $NO_2$ and $SO_2$ DSCDs are quite different in these two types of wind directions. When the wind is parallel to the observation direction (E and W), 34% of $NO_2$ DSCDs and 31% of $SO_2$ DSCDs are greater than $3.00 \times 10^{16}$ molec cm$^{-2}$ and $1.50 \times 10^{16}$ molec cm$^{-2}$, respectively. However, under the perpendicular direction (N and S), the occurrence of high DSCDs of $NO_2$ and $SO_2$ significantly decreased compared to parallel direction.

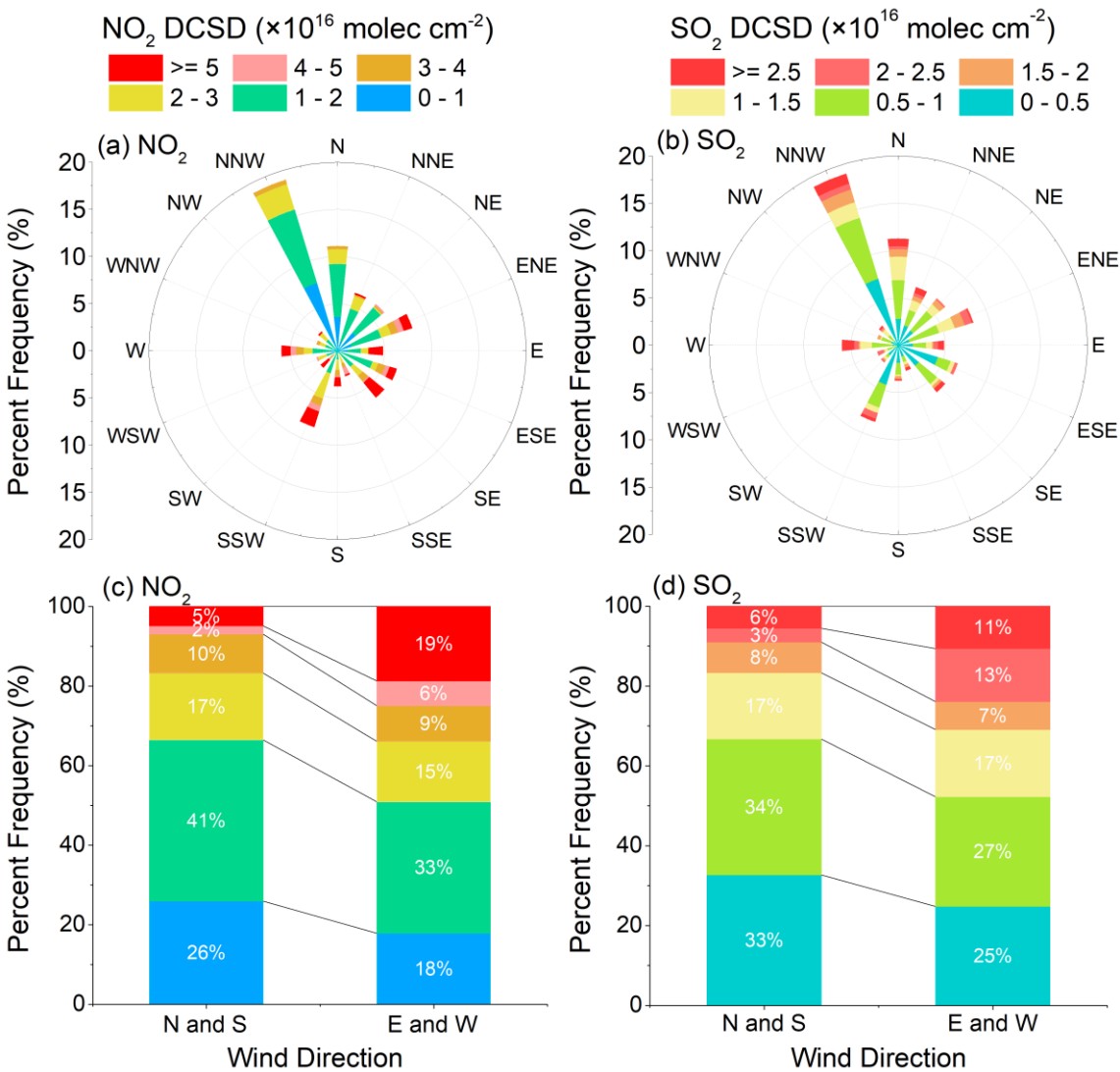

**Figure 8. The dependence of (a) and (c) of NO₂ and (b) and (d) of SO₂ DSCDs at elevation 5° on wind directions from January to March 2018.**

Based on the real-time photos taken by the instrument, we have counted the ship density manually to discuss its relationship with observed DSCDs of $NO_2$ and $SO_2$ at elevation 5°, as shown in Figure 9. It is obvious that the hourly means of $NO_2$ and

SO$_2$ DSCDs show an upward trend as the ship density increases. Since the fuel used by the ship is inconsistent, and the wind speed and direction also affect the DSCDs, it is difficult to find the clear linear relationship between hourly data of ship density and DSCDs in this busy inland waterway environment. From the prospective of statics, the averaged NO$_2$ and SO$_2$ DSCDs of binned ship density group (the hollow squares in the middle of the box) shows a strong positive correlation with the ship density, showing the correlation coefficient R of 0.86 and 0.97, respectively. The relatively higher R of SO$_2$ also suggests that the main impacts of SO$_2$ from ship emission source over there, however, more complex sources of NO$_2$ nearby.

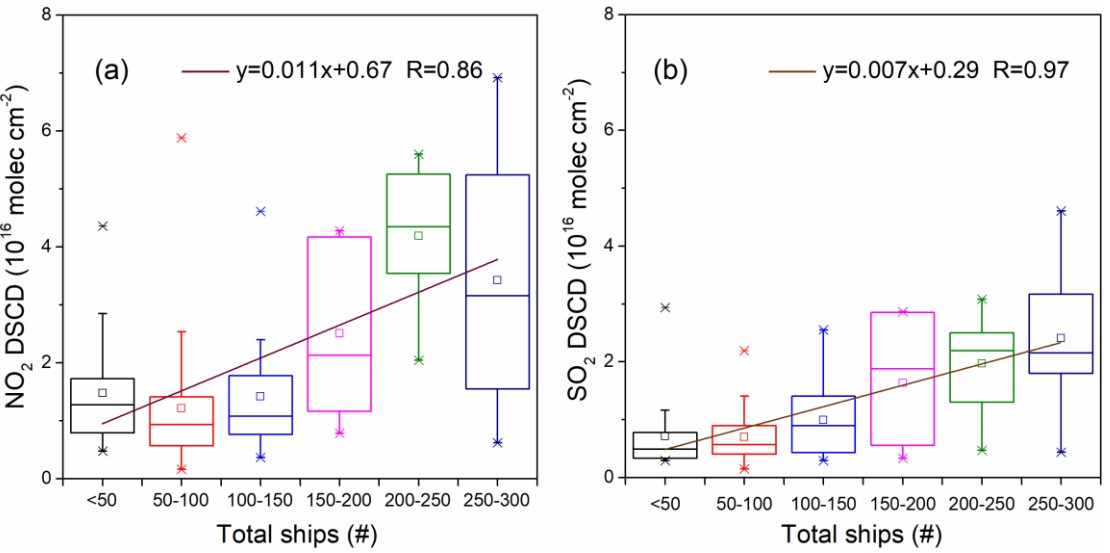

**Figure 9. Relationship between DSCDs of (a) NO₂ and (b) SO₂ at elevation 5° and ship density.** The hollow squares in the middle of the box represent the mean value, and the solid lines in the middle represent the median. Whiskers extend from each end of the box to the internal and external limits. "-" represents the maximum and minimum, and "×" are 1% and 99% quantiles. The upper and lower edges of the box are 25% and 75% quantiles, respectively.

In general, it can be concluded from the continuous several months measurements at Wusong MSB site that the ship density and meteorological conditions are the two major factors influencing the observed NO$_2$ and SO$_2$ levels in this typical inland waterways. For similar diffusion situations, the MAX-DOAS instrument can accurately detect the elevated pollutants concentrations with the increased number of ships. However, due to the busy ship lanes in front of the instrument, the MAX-DOAS instrument usually observes signal of pollutants in the plumes from multiple ships together, and the navigation speed of ships are relatively faster compared to the period of a completed scan measurement. So it is very hard to distinguish the single plume from the mixture. It is another shortcoming of this measurement that the MAX-DOAS measured NO$_2$ are considerably impacted by surrounding other emission sources, such as main roads and highways nearby, which contribute un-ignored amounts to the ambient NO$_2$. Since it is less practical for regulatory authorities to achieve fuel detection for each ship in this busy inland waterways, the remote sensing of MAX-DOAS measurement still offer the prospect of surveillance of ship emissions. Based on the legally sulfur content and ship activity data, the theoretical NO$_2$ an SO$_2$ concentration of plume

exhausted from the chimney can be calculated. By combining with the diffusion model of plume, the theoretical concentration of $SO_2$ on the light path of MAX-DOAS observation can be obtained. Afterwards, the suspicious ship using fuel with excess sulfur content can be identified.

### 3.3 Ocean-going ship emissions

Another shipping traffic scenario was also considered for the MAX-DOAS measurements at Yantian Port (114°29′ E, 22°56′ N), located on the east side of Shenzhen in the Pearl River Delta emission control zone (see Fig. 1). Yantian Port is one of the largest container ports in China and even in the world, which has 20 large deep-water berths with a quay length of 8,212 m and water depth alongside of 17.4 m, which is benefit to the ocean-going vessels with a length of more than 300 m docked. Unlike the measurement sites above, the distinct feature of shipping traffic in Yantian Port is the huge size of inbound and outbound vessels and the much less traffic density. The MAX-DOAS instrument was installed at the shore of the central operation zone of the Yantian Port from May 23, 2018. As can be seen in Fig. 1(d), the view direction of MAX-DOAS was pointed to the lanes in the eastward sea area. Due to lack of other emission sources in the front, the MAX-DOAS observation can easily capture the pollutants in single plume from the individual inbound and outbound ship, as manifested in Fig. 10.

Figure 10(a) and (b) presents the altitude dependence of observed $SO_2$ and $NO_2$ DSCDs around noontime on May 26, 2018. During the observational period, there were three apparent peaks of $SO_2$ and $NO_2$ DSCDs, i.e. 13:00, 13:30 and 14:10. The increases of both pollutants levels occurred simultaneously. For the first pulse around 13:00, the higher levels of $SO_2$ DSCDs are distributed above 10° elevation, whereas the strong signals of $NO_2$ are concentrated below elevation angle of 5°. This can be explained by the fact that the container ocean-going vessel and tugboat behave differently in emission and operation. Fig. 10(c) proved that there a large container ship is outbound at 12:55 with assistance of two tugs. It is obvious that height of outlet is very high for large container ships, but quite low for the tugboat. Since the tugboat are usually operated in the port area, its fuel usage always obey the regulations of ECA and shows high quality. Thus, stronger $SO_2$ signal appeared at high altitude due to the container ship emission, while $NO_2$ hotspots closed to the sea surface contributed by the tugboat emission. During the period around 13:30, both DSCDs of $SO_2$ and $NO_2$ were slightly increased and allocated below elevation 7°. According to the live photo in Fig. 10(d), there only a small container ship was passing through about 1 km away in front view of the instrument. Considering the distance between the ship and instrument, the height of exhaust outlet should be related to a lower elevation angle, where the corresponding strong signal of emitted pollutants are expected to be observed. Additionally, obvious $SO_2$ and $NO_2$ signals were found around 14:10, during which high $SO_2$ were distributed among elevation angle 10° to 15°, but strong signals of $SO_2$ and $NO_2$ were both found near the sea surface. However, no ship was captured by the live photos. It can be inferred from the AIS information that the observed signal of plume was dispersed from the ship in another lane instead of this in the front view of the instrument. Besides, the AIS information also confirmed that there are no other ship emissions disturbances for the two earlier measurements.

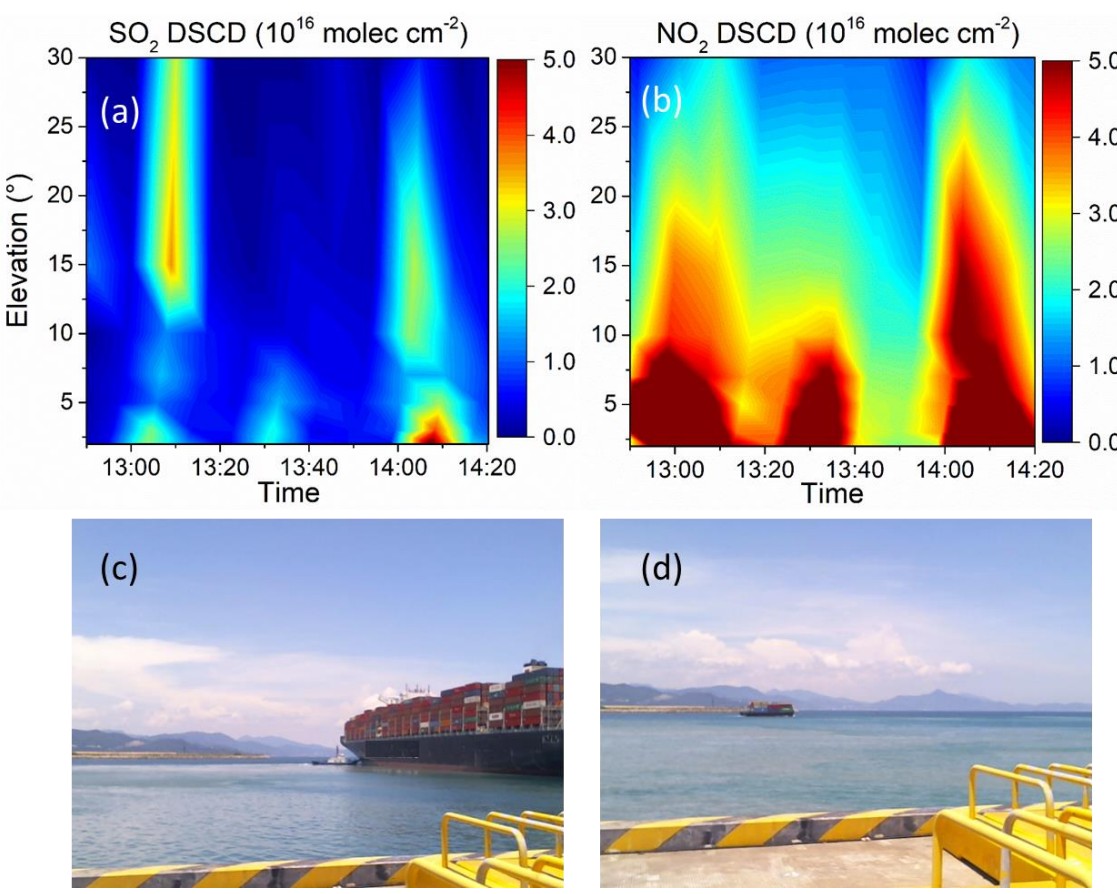


**Figure 10. Measured DSCDs of (a) SO₂ and (b) NO₂ during 12:55~14:20 and live photos taken by the camera at (c) 12:56 and (d) 13:22 on May 26, 2018.**

In general, the characteristics of observed SO₂ and NO₂ DSCDs distributions on height are to some extent related to the ship
size, its distance from the instrument and operational status, as well as the atmospheric stability. Based on the example discussion above, the MAX-DOAS measurement in Yantian port can detect the pollutants in single plume from individual ship and provide the information about vertical distribution of pollutants for conditions of low ship traffic volume. Considering the large discrepancies of SO₂ signals in altitudes, we try to analyze the detected plumes in more detail and obtain the representative observation elevation. According to the live photos, a large container ship entered the field of view at 09:51 on June 22, 2018,
which moved very slowly and emitted a distinct black smoke. Figure 11 shows the distribution of SO₂ DSCDs in plumes at different elevation angels. The SO₂ DSCDs peaked at $8.17 \times 10^{16}$ molec cm$^{-2}$ between elevation angles of 5° and 7°, and decreased with height. It was found that the DSCDs observed at elevation angle 7° are suitable to represent the peak concentrations in the plumes considering the chimney height of ship and its horizontal distance from MAX-DOAS instrument.

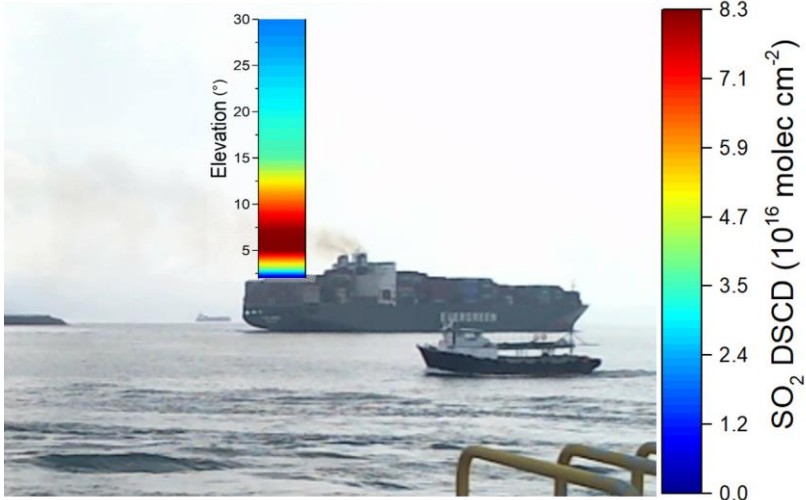

**Figure 11. A typical distribution of SO₂ DSCDs in the smoke plume of ship on June 22, 2018**

Instead of the full elevations scanning, with the temporal high resolved measurements (60 s) at single 7° elevation, it is possible to resolve individually the plume signal of passing ships. Thus, the diurnal profiles of $SO_2$ and $NO_2$ DSCDs at 7° elevation on June 26, 2018 were further investigated as presented in Fig. 12. There multiple peaks of $SO_2$ and $NO_2$, with the highest DSCDs

of $SO_2$ and $NO_2$ exceeding $6.00 \times 10^{16}$ molec cm$^{-2}$, occurred due to the emissions of the occasional passing ships. By applying the mathematical method, a baseline (the blue dotted line in Fig. 12) can be extracted from the DSCDs trend lines (the black solid line in Fig. 12). The baseline represents the diurnal variations of DSCDs, mostly due to the change of light path caused by solar zenith angle and the background emissions. Finally, it can be found that seven synchronous peaks of $SO_2$ and $NO_2$ levels higher than $2.00 \times 10^{16}$ molec cm$^{-2}$ are present in the trend line (the red solid line in Fig. 12). Validated by the live photos

of the instrument and the AIS information, these kind of sharp increased concentration of pollutants are originated from the ship plumes passed by. It suggests the high sensitivity of MAX-DOAS measurements to the change of $SO_2$ and $NO_2$ contents in the atmosphere. The increases of pollutants levels lasted from 10 min to half an hour, which is related to the durations of the ship movement in the field of view.

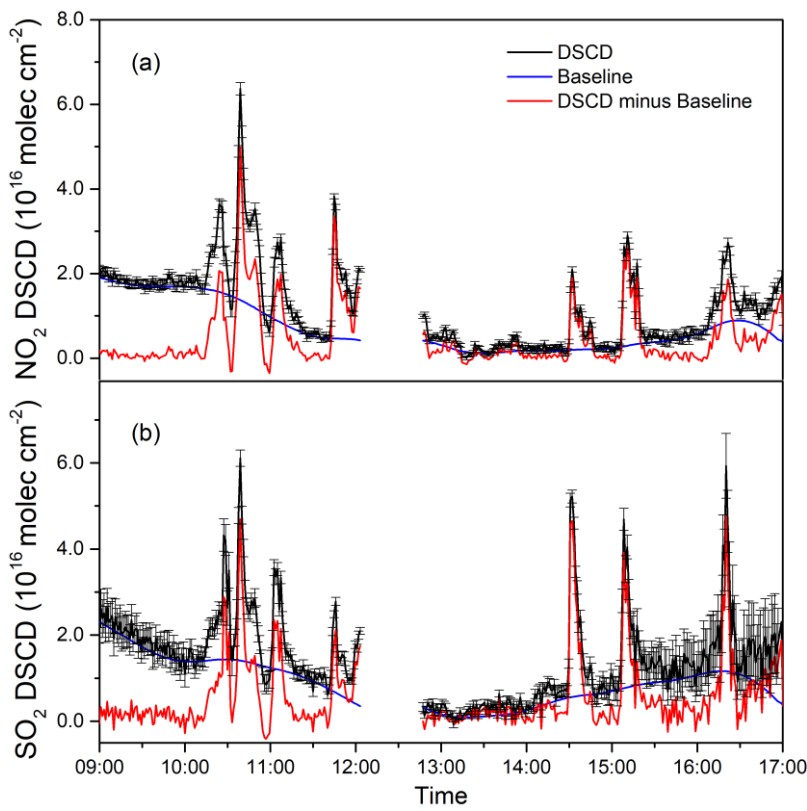


**Figure 12. Diurnal variations of DSCDs of (a) NO₂ and (b) SO₂ measured at 7° elevation angle on 26 June 2018.**

Moreover, it should be noticed that the amplitude of each peak varied differently, which implies that the DSCDs ratio of $SO_2$ to $NO_2$ for each peak may reveal the emission information of fuel sulfur content of individual vessels (Seyler et al., 2017;

Mellqvist et al., 2017). However, it is important to note that the $NO_2$ is formed by the reaction of NO and $O_3$ in the plume, so the $SO_2/NO_2$ ratio depends on the age of the plume to a certain extend. The ambient $O_3$ between 08:00 and 17:00 was averaged at 63.7 ppb in Yantian during the campaign. Considering the abundance of ozone, the NO emitted by the ship will react with $O_3$ rapidly to form $NO_2$ within a few minutes or even faster (Seyler et al., 2017). In addition, the conversion between NO and $NO_2$ is very fast and maintains a dynamic balance with sunlight during the daytime considering the photolysis of $NO_2$ (Singh

et al., 1987). Therefore, the $SO_2/NO_2$ ratio in the observed plume could be in a stable conditions and less impacted by the fresh emitted NO. Thus, the linear regression analysis between $SO_2$ and $NO_2$ DSCDs were performed to infer the fuel sulfur content.

Figure 13 presents the analysis results of 24 different vessels. The strong correlation relationship between $SO_2$ and $NO_2$ DSCDs are the obvious evidence of the significant homologies of emission sources between $SO_2$ and $NO_2$. Nevertheless, the slope of

different vessels highly changed from 0.28 to 2.90, indicating the diversity of the $SO_2$ emission intensity in the ship plumes. In general, the $SO_2$ emission are directly related to the fuel sulfur content and engine operation status of the ships according to

the emission model estimation, e.g. the power, activity time and the speed of ship (Fan et al., 2016; Zhang et al., 2018b). The outbound vessels usually leave from the shore slowly with the help of tugboat, and then speed up sailing into the sea. It calls the main engine to power the navigation during this process, and the fuel used by the main engine has a higher sulfur content than the auxiliary machine. In contrast, the main engine of vessel is usually shut down during the inbound process. Therefore, the ratio of $SO_2$/$NO_2$ DSCDs in the plume emitted by the outbound vessel could be higher than the inbound one.

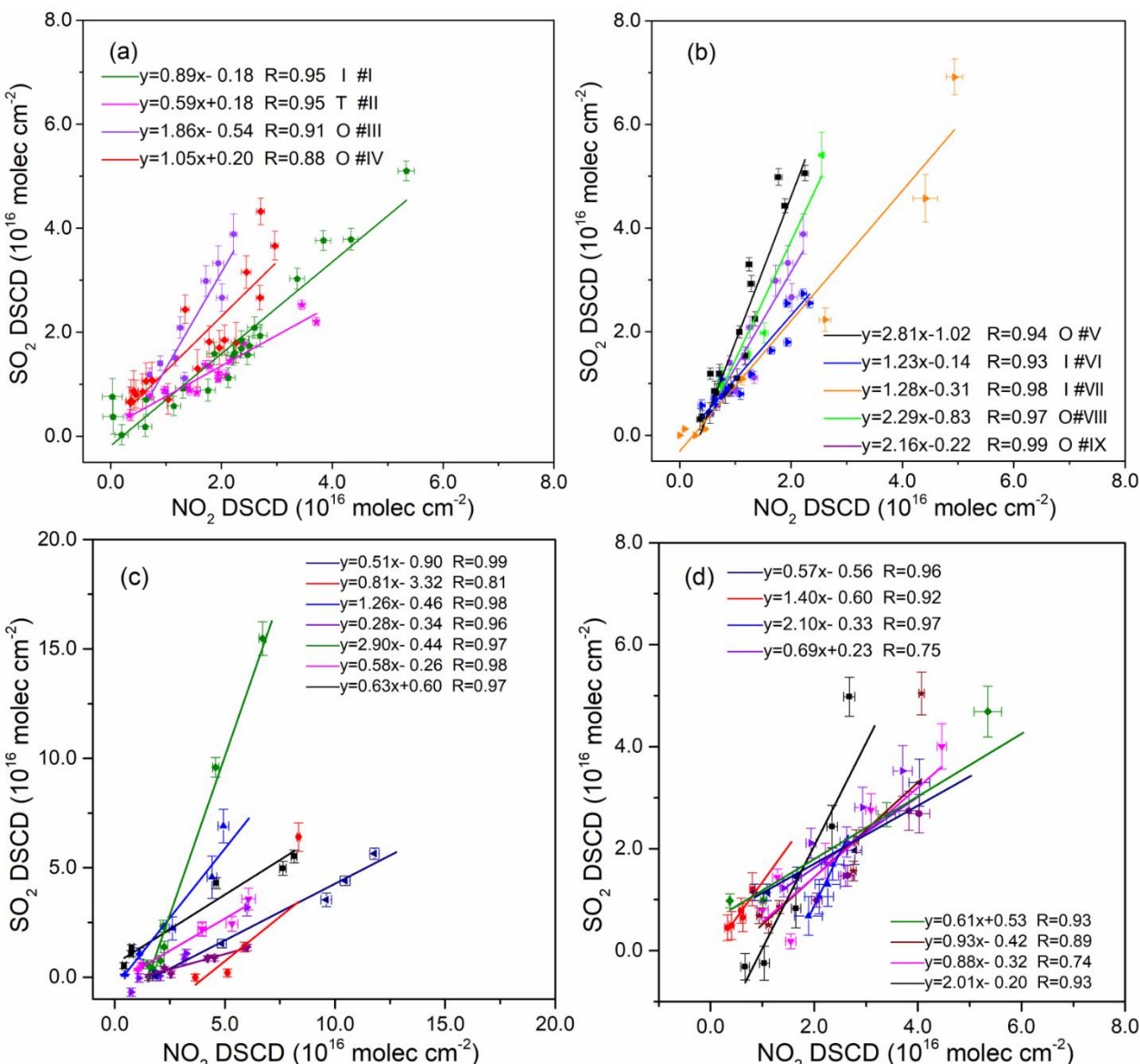

**Figure 13. The relationship between $SO_2$ and $NO_2$ emitted by several typical vessels.**

During the MAX-DOAS observation, we have also carried out some fuel sample analysis, investigation on the activity data and engine parameters of these vessels, among which five of them are the vessels in Fig. 12. Therefore, we indicated the different status of nine vessels in Fig. 14, along with the information on the rated power of engine and fuel sulfur content individually, in which the inbound and outbound shows the rated power of main and auxiliary engine, respectively. The vessel #II is a tugboat operated in the port area, which uses the fuel with lowest sulfur content of 0.001% and shows minimum ratio of $SO_2/NO_2$ in DSCDs. Furthermore, the inbound vessel #I, #VI and #VII (indicated by diamond in Fig. 14) have switched off the main engine when it arrived in the front of the MAX-DOAS instrument, moved under the towing of tugboat and docked by inertia finally. Additionally, the sulfur content of fuel are much lower in auxiliary engine than main engine. So $SO_2/NO_2$ ratios of inbound vessels is much lower than that of outbound vessels. The other vessels, indicated by circle in Fig. 14, are all in outbound. Under the launch of main engine and high sulfur content in fuel, these vessels exhibited relatively higher ratios of $SO_2/NO_2$ over 2.0, except for vessel #IV. Due to the usage of much more cleaner fuel with sulfur content of 1.28%, the vessel #IV presented the lowest ratio of $SO_2/NO_2$ among all the outbound vessels. Compared to vessel #III with similar rated power of engine, it can be observed that the ratio of $SO_2/NO_2$ in the plume increased with the increase of fuel sulfur content for vessels. This phenomenon is also applicable to cases of vessel #V and #VIII. It is worth to note that the circle of outbound cargo #IX is deviated from others, which has very low rated power but very high ratio of $SO_2/NO_2 > 2.0$. So it can be recognized as a suspicious ship using fuel with sulfur content exceeding the regular limit.

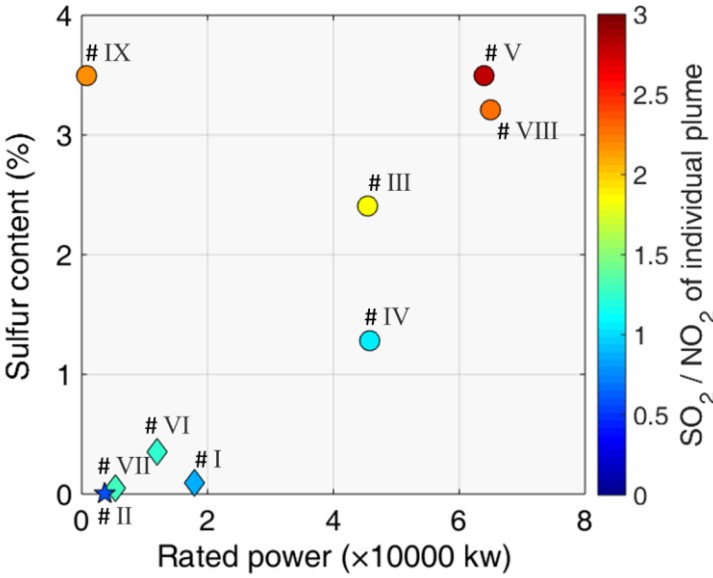

**Figure 14. The relationship between the ratio of SO₂/NO₂ obtained by linear regression, the fuel sulfur content and engine rated power of all nine vessels. Tugboat, inbound and outbound vessels are represented by pentagram, diamond and circles, respectively.**


Basically, the ratios of $SO_2/NO_2$ in the plume discharged from the inbound vessel and the tugboat are usually lower than 1.5 for normal condition, which is much smaller than that of outbound vessels using high rated power engine and high sulfur content fuel. For outbound vessels, the ratios of $SO_2/NO_2$ are more related to the fuel sulfur content. The irregular observed ratio of $SO_2/NO_2$ can tag the vessel not obeyed to the sulfur content limitation. Therefore, the MAX-DOAS measurement
provides a promising technology for compliance monitoring of fuel sulfur content by investigating the ratio of $SO_2/NO_2$ in the plume and the more accurate estimation with load factor and emission factor for the actual operation. Besides, the statistics of $SO_2/NO_2$ ratios in discharges were performed for 55 ships during the observation. The frequency distribution of the slope of $SO_2/NO_2$ is shown in Figure 15. It shows that the values of $SO_2/NO_2$ were mostly distributed less 1.5, which occupied about 72.7%. Ships with the ratio of $SO_2/NO_2$ between 0.6 and 0.9 shared the highest proportion. It indicates that most of the fuel
used by ships in Yantian Port could be qualified. However, there are still some ships may use non-compliant fuel, because the ships with a value of $SO_2/NO_2$ greater than 1.5 account for 27%.

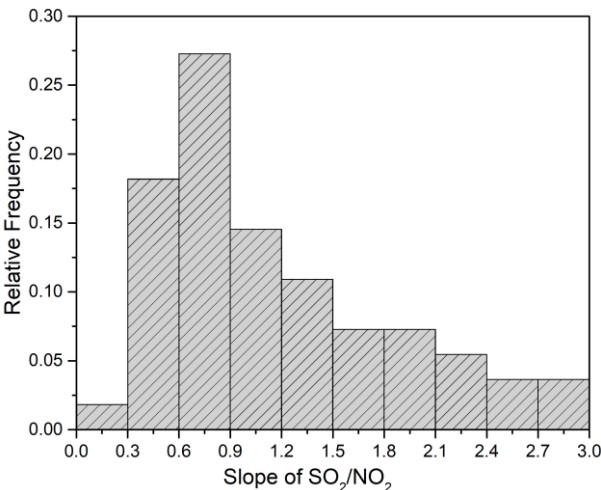

**Figure 15. Frequency distribution of the slope of $SO_2/NO_2$ from samples of 55 vessels.**

## Conclusion


In this study, we performed the MAX-DOAS measurements observe ship emissions of $SO_2$ and $NO_2$ in Shanghai and Shenzhen, China for three different typical ship traffic conditions. At Waigaoqiao container terminal in Shanghai, the $SO_2$ and $NO_2$ exhausted by ship at berth can be easily identified for the locations and intensity of emission from the 2-dimensional MAX-DOAS observation. At the inland waterway area of Wusong Wharf it is difficult to determine the single ship emissions due to
the dense traffic volume and complex background environment. The long-term MAX-DOAS measurements shows that the changes of $SO_2$ and $NO_2$ are correlated to ship traffic density at stable and unstable atmospheric conditions. However, better dispersion under unstable atmospheric condition are favorable for the decrease of pollutants levels. For open sea waters in

Yantian deep water port, the DSCDs of $SO_2$ and $NO_2$ measured by MAX-DOAS are highly sensitive to the emitted plumes of vessels passing through in front of the shore-based instrument, which shows the significant increase of pollutants concentrations and 10-30 min duration of the emission signals. Considering the distance and size of the vessels, the DSCDs observed at elevation angle 7° are the hotspots of the concentration in altitude, and further selected to investigate the fuel sulfur content. According to the linear regression of $SO_2$ and $NO_2$ DSCDs, the ratio of $SO_2/NO_2$ are found to be very helpful to infer the levels of sulfur emission. Combined the fuel sample analysis and inquiry of the vessel data, the $SO_2/NO_2$ ratio in the plume are usually lower than 1.5 for the inbound vessel and the tugboat, whereas is much smaller than that of other vessels. The abnormal high ratio of $SO_2/NO_2$ in the plume usually implies the vessel could not be in compliance with the sulfur content limitation.

In summary, the advantages of optical remote sensing and mature for $SO_2$ and $NO_2$ detection are beneficial to MAX-DOAS measurement for the ship emission. These applications at different ship traffic scenarios demonstrated the feasibility of shore-based MAX-DOAS to observe the emitted $SO_2$ and $NO_2$ from vessels docked at berth, navigation in the lanes, inbound and outbound operations. Nevertheless, the main ship emitted pollutants of NO and $CO_2$ cannot be monitored due to the limitation of the observed wavelength range. Since MAX-DOAS uses solar scattered light as the source, it cannot be measured at night when there is no sunlight, and there is a large error during twilight and rainy observations. For the prospects, the combination of MAX-DOAS remote sensing of ship plumes and the estimation on emissions with theoretical fuel sulfur contents and actual operation data will provide the promising approach for surveillance in the future.

**Data availability.** Data are available for scientific purposes upon request to the corresponding authors.

**Author contributions.** YC, SW and BZ designed and implemented the research, as well as prepared the manuscript; JZ, YG and RZ contributed to the MAX-DOAS measurements at different sites; YC and SW carried out the MAX-DOAS retrieval and analysis combined with other auxiliary data; YL, YZ, YQ and WM provided constructive comments and support for the ship emissions research of this study.

**Competing interests.** The authors declare that they have no conflict of interest.

**Acknowledgements**

This research was supported by grants from National Key Research and Development Program of China (2016YFC0200401, 2017YFC0210002), National Natural Science Foundation of China (41775113, 21777026, 21677038), Shanghai Pujiang Talent Program (17PJC015), Pudong Science and Technology committee of Shanghai (PKJ2018-C05). We would like to thank

Shenzhen Maritime Safety Administration, Wusong and Pudong Maritime Safety Bureau of Shanghai for the coordination of field measurement, respectively.

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
