# Peer review of "Surveillance of SO2 and NO2 from ship emissions by MAX-DOAS measurements and implication to compliance of fuel sulfur content"

_Atmospheric Chemistry and Physics, 2019_

## Referee Comment (RC1) · Andreas Weigelt (Referee) · 31 May 2019

Overview: The manuscript present results from half a year of MAX-DOAS ship emission measurements at two different regions in the Chinese emission control area. With some examples the authors show the potential of the DOAS measurement technique to monitor ship emissions in general and of individual ships in transit. This is of high interest for the scientific community, dealing with ship emission measurements. However, the database the authors use for their conclusions is weak. Some analyses have to be improved. So mayor revisions are needed to consider this manuscript for publication in ACP.

[Figure]

General comments:

In the introduction the authors mention SO2 and NO2 to be the "main pollutants of ship emissions" (line 31-32). As the fuel consists of ca. 87% carbon, CO2 is by far the main pollutant in ship plumes. Beside CO2, Nitrogen monoxide (NO) is the second dominant pollutant in ship plumes. NOx is emitted mainly as NO and not as NO2. The NO2 is formed in the plume when the plume ages (NO + O3 –> NO2 + O2). Our measurements within the German ship emission monitoring network usually shows NO/NO2 ratios of above five when the plume age is less than 5 minutes. With increasing time the NO/NO2 ratio decreases to below 1 (age > 15 minutes). The MAX-DOAS instrument cannot measure CO2 and NO. Nevertheless, in the introduction the authors should consider CO2 and NO as the main pollutants of ships. They should also discuss influence of the NO –> NO2 transformation inside the plume to their measurements.

At the end of the introduction (Line 97-100) and in section 3.3 it is stated that the measurements can be used to estimate a fuel Sulphur content (FSC) from the SO2/NO2 ratio. As already mentioned in the first general comment the amount of NO2 and therefore the SO2/NO2 ratio strongly depends on the age of the plume. Directly at the stack the SO2/NO2 ratio should be highest and then decreasing with increasing time. The authors should consider this in their description and calculations.

Results presented in section 3.1 do represent only one measurement at one day. If the authors have "more than a dozen cycles" (line 220-221), why they don't show an average of all cycles. The presented cycle is a snapshot and might be not representative to draw conclusions. Well, even an average over one afternoon does represent only a snapshot of the situation. The authors should consider this in their discussion or should skip this section.

Also in Section 3.2 the authors discuss only data from two selected day. To make more general conclusions the authors have to average a longer time period (if possible all Wusong measurements).

Again in section 3.3 the conclusions are based on very little data points (only nine!), which is critical to make general statements. So for sound statements much more data have to be analyzed.

At least in the Conclusion the authors should discuss not only the advantages but also the limitations of the MAX-DOAS method, which are: - no NO and CO2 measurements (which are the main components inside the plume) - no measurement during twilight and night.

Specific comments

Line 15: It is confusing that the authors mention that the measurements took place in Shanghai and Shenzhen and the next sentence starts with "These three typical measurement sites..." In Section 2.2 it becomes clear that the measurements were performed at three sites in two regions –> please clarify in the abstract.

Line 37: consider also CO2 (see first general comment)

Line 42-43: what kind of important role do the ship emitted pollutants play in air quality, human health and climate? Political regulations, monitoring, enforcement, ...?

Line 53-54: The authors have to distinguish different limits in different ECAs: So the maximum fuel Sulphur Continent (FSC) in ECAs at the US coast and Europe (whole Baltic- and North Sea) is 0.10% S m/m. Inside the Chinese ECA it is 0.50 % S m/m. By 2020 the maximum FSC is 0.50% S m/m all over the world (global Sulphur cap) which is not related to designated ECAs. As the use of exhaust gas treatment systems (Scrubber) is allowed as an alternative, the authors should mention this option, too.

Line 57: Are Scrubbers allowed in the Chinese ECA? If yes, this has to men mentioned here, too.

Line 64: "... usually fast detecting..." –> What is fast (minutes, hours, days,...)?

Line 69: the authors should give a reference (e.g. Kattner et al. 2015, or Seyler et al.

2017)

Figure 2: Do the authors have the copyright for the satellite pictures in Fig. b-d? If not, they have to cite the source.

Figure 3: It looks like graph (b) and (d) are mixed up, because the y-scales do not match the y-scales of graph (a) and (c). Can the authors confirm this? If not, the authors should comment in the corresponding discussion (line 170-175) why the scales are different

Figure 3 (line 180-181): Wouldn't it be better co compare measurements with same elevation angle to minimize differences due to different optical length inside the planetary boundary layer (PBL)?

Line 190: What is located opposite the berth? Any industry which might emit NOx or SO2?

Line 196: Why only 10° angle for the reference spectrum? At 10° the way thru the PBL is still long and therefore might be influenced by emissions. Why the authors did not measure at 90° or at least at 65° like in Wusong?

Figure 4: In my opinion this figure is not necessary to explain the measurements. So if the authors want to save some space, they could skip this figure.

Line 208-210: please explain with compass direction (e.g. shift to north-west) to compare it to the given wind direction. What dose "MAINLY came from the south" mean? Here only one measurement (a 15 min scan) is discussed. Dose the wind direction changed during this 15 minute scan?

Line 220-221: If the authors have "more than a dozen cycles" why they don't show an average of all cycles. The presented cycle is a snapshot and might be not representative to draw conclusions.

Line 253: ships in navigation or maneuver? Do the authors can exclude emissions

from the industry behind the river (visible in Fig 2 (c)) as a source?

Line 256-268: Why the authors do only check for the influence of wind speed? I would expect a much bigger correlation with wind direction because the optical length inside the polluted air and therefore the response signal is probably increased when the wind transports the polluted air towards the DOAS. On the contrary the optical length inside the polluted air and therefore the response signal is probably less when the wind transports the polluted air perpendicular to the DOAS (out of the field of view). The authors should check the wind direction dependency for all data. Why constant high wind speed (5-6 m/s is not high) unstable atmospheric conditions? As stated above, in my opinion the wind direction is of high importance as well.

Line 263-264: The close relation of SO2 and NO2 signal to the flow of ships (better use "ship density) is not clear on March 09 12:00-14:00. At this time the ship density is constant but SO2 and NO2 decrease.

Figure 7 and corresponding discussion: If the authors want to correlate SO2 and NO2 with traffic density and meteorological conditions, they should use all data and not only data from two days. They should create scatter plots (e.g. ship density on x and SO2 on y-axis) to find correlation coefficients. As the authors give a scan time of 7 minutes for the Wusong measurements (Table 1), each box-whisker-plot is based on only 4 measurements. This is not valid for this kind of plots. If the authors use all azimuth angles in one box, this is also not valid, because of different optical length and therefore not comparable measurement conditions.

Line 276-279: This conclusion might be true, but has to be proved not only by a snapshot but by an average as indicated in the comment above.

Line 282-283: To support the conclusion that the ships are the main source for NO2, the authors should roughly estimate the influence of the surrounding emission sources (especially the main roads and highway, because this could be a significant NO2 source).

Line 285-287: The meaning of this sentence in not clear. Do the authors mean that they want to use the complicated MAX-DOAS inland waterway measurements to check ships for compliance with fuel Sulphur regulations? It is not clear how the authors want to compare a theoretical SO2 emission (which unit?) with the MAX-DOAS measurement result (column density). This has to be explained in more detail. In my opinion comparing only SO2 from theoretical emissions to DOAS measurements does not work, because from the DOAS measurements you don't know whether you measure in line or perpendicular to the plume.

Line 292: largest container ports related to what? China, Asia, World?

Line 305-308: Why the big container vessel does not show any NO2 signal but the tug do so? The container vessel should emit even more NOx than the two tugs. Please discuss.

Line 314-316: This sentence put a question mark onto the measured plumes at 13:00 and 13:30. So the authors have to state that for the two earlier measurements no other ship could have caused the high signals (e.g. based on AIS analysis). Line 323: How the emissions are related to the operational status (was not discussed before)?

Line 326-327: "we try to further detailed SO2 emissions from the measurement" –> the meaning is not clear. Do the authors want to say that they try to analyze detected plumes in more detail?

Line 339: Which mathematical method? How the baseline is calculated (e.g. running mean or median; which averaging interval?)?

Line 351-355: The use of fuels with different Sulphur content is one possible explanation for the observation of different SO2/NO2 ratios. The age of the plume and the direct NO2 emissions is another plausible explanation. As already mentioned in the first general comment, ships mainly emit NO but not NO2. Therefore, directly at the stack the SO2-NO2 ratio is highest. When the plume ages NO2 is formed and the

SO2-NO2 ratio decreases. NO2 emissions are dependent on the kind of engine and the burning temperature of the engine as well. The authors have to consider this in their discussion.

Line 359-361: Is the main engine really operated with fuel with higher Sulphur content than the auxiliary engine? Do the authors have a source for this statement? I thought inside the Chinese ECA the maximum allowed fuel Sulphur content is equal to that allowed at berth.

Line 381: Do the authors mean vessel #IX instead of cargo #IV? What about vessel #V and #VIII? Are they allowed to use fuels with Sulphur content above 3% inside the Chinese ECA? I thought the limit is 0.5%

Line 391-394: from the SO2-NO2 ratio displayed in Fig. 12 it is not obvious which is the "irregular observed ratio". I do agree that if the SO2-NO2 ratio is above 1.5 this is an indication for the use of fuel with high Sulphur content. But from this point of view vessel III, V, VIII, and IX should be indicated as non-compliant. Is it possible to estimate a kind of detection limit for the observation of non-compliant vessels? Is it possible to distinguish between 0.4 (compliant) and 0.8 (non-compliant)? This would address also the conclusion (Line 410-412).

Technical corrections

Line 15: ships instead of ship

Line 22: ad "the" in front of SO2/NO2. . .

Line 26: Combining instead of Combined

Line 27: What is meant with "logical Sulphur content"?

Line 28: "more accurate way " –> than what?

Line 44: ad "of" in front of "kilometers"; remove "the" in front of "most"

Line 53: areas instead of zones

Line 57: should or must?

Line 85: close instead of closed

Line 95: ad "the" in front of instrument

Line 105-106: "...and stored in form of spectrum" –> It is not clear what is meant. The MAX DOAS instrument records spectra which represents the intensity of scattered sun light at different wave length (please give the scan interval). For each viewing direction and measurement interval a separate spectrum is recorded

Table 1: The measurement sites name and locations should start in the same line the operations AZ starts. At the moment it is little bit confusing that the line where the AZ is given starts above the site names.

Table 1: The authors should give the number of scan cycles, as well. As the measurements in Waigaoqiao last less than one day and one can takes 15 minutes I think there are not many scans available for analysis.

Line 186: ad "of" in front of "more than"

Line 189: ad "of" in front of "ships at"

Line 194: remove "can" in front of "covers about"

Line 204: increase instead of increases

Line 242: write "Beside of ocean-going ships, inland waterway vessels also contribute significantly to the amount of ship emissions..." instead of "Besides ocean-going ship emissions, inland waterway vessels also contributed significantly to the ship emissions..." –> not the emissions, but the ships are ocean-going :-)

Line 245: close instead of closed

Line 246-247: This sentence is not clear. I suggest to split it.

Line 251: ad a space (direction of)

Line 252: lane instead of lanes

Line 256: use singular instead of plural: impact of ship traffic; measurement data

Line 273: what do the bars and the stars do represent?

Line 291: see instead of See

Line 299: remove "orderly" in front of "inbound"

Line 300: 2018 instead of 2019; two dots at the end

Line 302: remove "were" in front of "occurred"

Line 310: was instead of were

Line 323: "more or less" –>please be more precise!

Line 325: "for" instead of "with the"

Line 330: "was" instead of "can be"

Line 331: "stand for the peak concentration" –>do the authors mean "represent the peak concentration"?

Line 337-338: It is difficult to get the meaning of this sentence. Do the authors want to say that with temporal high resolved measurements (60 sec) it was possible to resolve individually the plume signals of passing ships? Please rephrase.

Line 340: Comma after DSCDs

Line 342: "are present" behind cm-2

Line 345: remove "In addition" in front of "the increases of pollutants"

Line 368: "on" instead of "about"

Line 374: "is" instead of "are"

Line 379: ad "the" in front of "plume"

Line 380: "increase" instead of "growth"

Line 381: "to note" instead of "noted"; "circle instead of "dot"

Line 390: ad "high" in front of "sulfur content"

Line 393: "compliance monitoring" instead of "compliance"

Line 397: write "In this study we performed MAX-DOAS measurements to observe ship emissions of SO2 and NO2 in Shanghai…" instead of "In this study, we have performed the MAX-DOAS measurements observe the ship emissions of SO2 and NO2 in Shanghai…"

Line 400: delete comma

Line 402: better "… are correlated to ship traffic density at stable and unstable atmospheric…"

---

## Referee Comment (RC2) · Anonymous Referee #2 · 1 Jun 2019

The paper presented by Cheng et al. has reported the shore-based MAX-DOAS measurements of ship emitted SO2 and NO2 under three different conditions in China's ship emission control area (ECA), i.e. ship docked at berth, navigation in the inland waterway and inbound/outbound in the deep water port. Although the detection of SO2 and NO2 by MAX-DOAS has been developed for many years, the employments for ship emission surveillance are an interesting application of the MAX-DOAS technique. I think the manuscript fits to the scope of ACP, especially for this special issue. I recommend publication after the authors addressed the following comments.

Major concerns:

[Figure]

The authors use the SO2 and NO2 DSCDs measured at different elevations for the evaluation of ship emissions. However, the vertical distribution of background SO2 and NO2 are quite different. It is not clear that how do the authors separate the ship emissions of SO2 and NO2 signal from the background? This information has to be supplemented in section 2.

The sectioning of section 2 is not very logical. I suggest the authors follow the order of "instrument", "spectral retrieval" and "ship emissions identification".

In section 2.3, the SO2 and NO2 DSCDs are retrieved at different spectral ranges. How do the authors compensate the effect of wavelength dependency? If it is not considered in the retrieval, an error analysis is required.

Sect. 3.1, In the 2D scanning, the authors used the reference spectrum measured at azimuth angle of 10, however, it can be seen from Fig. 2(b) that this direction are still pointing to the berth. How to confirm the impacts of ship emission in the reference spectrum has been excluded? Alternatively, how to evaluate the uncertainties on the absolute value of DSCDs due to this?

Both in Sect. 3.1 and 3.3, the authors used the mathematic method to the slowly change of DSCDs in temporal pattern. I think the author should introduce something more about why this method can be used here? And the basic principle? In line 340, how to prove that the baseline represents the diurnal variations of DSCDs mostly due to the change of light path caused by solar zenith angle and the background emissions?

The authors have mentioned that it is difficult to distinguish the single ship plume. How do the authors derive the emissions from different vessels (Figure 11)? How the data are filtered? What is the error?

Minor comments:

What is the typical error of the measurements? Please put the error bars on figure 6, 10 and 11.
Figure 11 is very busy. It is difficult to see the differences between each species. Maybe the authors can separate it into 2 to 3 subplots. More detailed caption is required.

Technical corrections:

Line 17, "berth" to "berths"

Line 105, "instruments" to "instrument", "observe" to "observes"

Line 111, "less trace gas absorptions"

Line 112, what is the slope column concentration? It should be the slant column density.

Line 122, "impacted by"

Line 174, "unqualified NO2 and SO2 DSCDs" to "unsatisfied spectral fitting", and "fitting results" to "DSCDs results".

Table 1 title, "operative" to "operation"; in the line of "Yantian", "Smaller" with unnecessary capital letter.

Table 2, whether the O4 absorption was included in the SO2 fitting range? What's meaning of symbol "–" standing for here?

Line 191, "multiple berth" to "multiple berths"

Line 226, "the residual after background subtraction"

Line 247, "boxes serving"?

Line 268, "around 5 m•s-1 on March 9"

Line 277, "impacting" to "influencing"

Line 292 and 293, "meters" can be shorten as "m"

Line 300, there are two dots in the end of the sentence. Please delete one.

[Figure]

Fig. 11, I suggest to also indicate the inbound and outbound status of the vessels to easily exam the relationship with slope.

Line 371, SO2-to-NO2 > SO2/NO2, also in the rest of the manuscript.

Line 381, IV or IX ?

Line 405, where is the 2-D DSCDs map at Yantian in manuscript?

Line 390 and 410, what the ratios of SO2/NO2 of inbound vessels and tugboat? Lower than 1.3 or 1.5? Please keep the consistency of description.

---

## Author Comment (AC1) · 25 Jul 2019

**Response to comments from reviewer #1**

We thank the reviewers for the constructive comments and suggestions, which are very positive on the scientific content of the manuscript. We have revised the manuscript appropriately and addressed all the reviewers' comments point-by-point for consideration as below. The remarks from the reviewers are shown in black, and our responses are shown in blue color. All the page and line numbers mentioned following are refer to the revised manuscript without change tracked.

Reviewer:

Overview: The manuscript present results from half a year of MAX-DOAS ship emission measurements at two different regions in the Chinese emission control area. With some examples the authors show the potential of the DOAS measurement technique to monitor ship emissions in general and of individual ships in transit. This is of high interest for the scientific community, dealing with ship emission measurements. However, the database the authors use for their conclusions is weak. Some analyses have to be improved. So mayor revisions are needed to consider this manuscript for publication in ACP.

General Comments:

1. In the introduction the authors mention SO2 and NO2 to be the "main pollutants of ship emissions" (line 31-32). As the fuel consists of ca. 87% carbon, CO2 is by far the main pollutant in ship plumes. Beside CO2, Nitrogen monoxide (NO) is the second dominant pollutant in ship plumes. NOx is emitted mainly as NO and not as NO2. The NO2 is formed in the plume when the plume ages (NO + O3 –> NO2 + O2). Our measurements within the German ship emission monitoring network usually shows NO/NO2 ratios of above five when the plume age is less than 5 minutes. With increasing time the NO/NO2 ratio decreases to below 1 (age > 15 minutes). The MAX-DOAS instrument cannot measure CO2 and NO. Nevertheless, in the introduction the authors should consider CO2 and NO as the main pollutants of ships. They should also discuss influence of the NO –> NO2 transformation inside the plume to their measurements.

R: We have not considered this point rigorously in previous manuscript. Since the limitation of the wavelength range of the instrument, $CO_2$ and NO cannot be detected in this study. Therefore, here we mainly focused on the $NO_2$ and $SO_2$ emitted by the ship, while the other two main pollutants of $CO_2$ and NO were not introduced in the Section 1. We have added the introduction about ship emitted NO and $CO_2$ and the NO/$NO_2$ ratio in different aged plume in the revised manuscript, as well as related references. Please refer to Line 37-38, 104, 396-402, and 472-473.

2. At the end of the introduction (Line 97-100) and in section 3.3 it is stated that the measurements can be used to estimate a fuel Sulphur content (FSC) from the SO2/NO2 ratio. As already mentioned in the first general comment the amount of NO2 and therefore the SO2/NO2 ratio strongly depends on the age of the plume. Directly at the

stack the SO2/NO2 ratio should be highest and then decreasing with increasing time. The authors should consider this in their description and calculations.

R: Thanks for the comments. We have not considered properly in the previous manuscript. NO can converted to $NO_2$ by reaction with $O_3$ in the atmosphere, and the rate of NO conversion to $NO_2$ is strongly affected by the concentration of $O_3$ (Han et al., 2011). The average value of $O_3$ between 08:00 and 17:00 in Yantian was 63.7 ppb during the campaign. Considering the abundance of ozone, the NO emitted by the ship will react with $O_3$ rapidly to form $NO_2$ within a few minutes or even faster (Seyler et al., 2017). In addition, $NO_2$ is photolyzed by UV radiation to release NO and oxygen radicals. In the collision reaction with $N_2$ or $O_2$, oxygen radicals react with oxygen molecules to reform $O_3$. So the conversion between NO and $NO_2$ is very fast and maintains a dynamic balance with sunlight during the daytime (Singh et al., 1987). It can be considered that the plume measured by MAX-DOAS was stable after the dynamic reaction.

Besides, there is a strong correlation between the DSCDs of $SO_2$ and $NO_2$ with the average value of R higher than 0.9 in this study. It proves that there is no significant change in the value of $NO/NO_2$ in the observed plume. Seyler et al. (2017) and Mellqvist et al. (2017) have used this relationship for ground-based and airborne DOAS measurements of ship plumes to distinguish ships with low (0.1 %) and high (1 %) fuel sulfur content. These experiments have proved that the low and high sulfur oil can be better distinguished based on $SO_2/NO_2$. (Seyler et al. 2017). The corresponding discussion has been added to the manuscript. Please refer to Line 396-402.

3. Results presented in section 3.1 do represent only one measurement at one day. If the authors have "more than a dozen cycles" (line 220-221), why they don't show an average of all cycles. The presented cycle is a snapshot and might be not representative to draw conclusions. Well, even an average over one afternoon does represent only a snapshot of the situation. The authors should consider this in their discussion or should skip this section.

R: There may be some misunderstandings due to the unclear description. The purpose of Section 3.1 is to investigate the 2-D distribution of retrieved $NO_2$ and $SO_2$ DSCDs using 2-D scanning measurement of MAX-DOAS. The hotspots of $NO_2$ and $SO_2$ are related to the azimuth of the berth where the ship is docked and the corresponding ship operation status. It is obviously that the 2-D distribution of $NO_2$ and $SO_2$ DSCDs changes with time, so that it seems unreasonable to show the average of all scan cycles, which may weaken the spatial distribution of hotspots. However, the results of the rest of the cycles were also shown in Figure 6. It can be seen that the concentration at a given azimuth and elevation is constantly changing throughout the day.

4. Also in Section 3.2 the authors discuss only data from two selected day. To make more general conclusions the authors have to average a longer time period (if possible all Wusong measurements).

R: Thanks for the suggestion. We have further analyzed the data measured at Wusong from January to March 2018. We have combined the hourly mean value of $NO_2$ and

SO₂ over three months with the wind direction and found that the average DSCDs of NO₂ and SO₂ is higher when the direction of wind is parallel to the observation direction. In addition, the DSCDs of NO₂ and SO₂ also shows a similar trend to ship density. Please refer to the responses to specific comments 17 and 19. This part of the content is also added to the manuscript. Please refer to Line 289-315.

5. Again in section 3.3 the conclusions are based on very little data points (only nine!), which is critical to make general statements. So for sound statements much more data have to be analyzed.

R: Due to the few samples of ship details we obtained on site, only 9 ships were discussed in the conclusion of Section 3.3. We have added more data analysis in Section 3.3. The Figure R1 shows the relationship between SO₂ and NO₂ DSCDs from multiple ships emission during the Yantian observation in June 2018. Figure R1 (c) and (d) are the supplementary analyzed ships but no related fuel information and operative status were available.

[Figure]

**Figure R1. The relationship between SO₂ and NO₂ emitted by several typical vessels, the letter "O" indicates the outbound vessels, "I" indicates the inbound vessels, and "T" indicates the tugboat.**

Besides, we have made a statistics for the values of $SO_2/NO_2$ discharged from 55 ships. The frequency distribution of the slope of $SO_2/NO_2$ is shown in Figure R2. It shows that the values of $SO_2/NO_2$ were mostly distributed between 0.0 and 1.5 with the proportion is about 72.7%. Ships with the value of $SO_2/NO_2$ between 0.6 and 0.9 have the highest proportion, appeared for 15 ships in total. It indicates that most of the fuel used by ships in Yantian Port could be qualified. But there are still some ships may use non-compliant fuel, because the ships with a value of $SO_2/NO_2$ greater than 1.5 account for 27%. This part of analysis has been added to the manuscript. Please refer to Line 404-415, 442-449, Figure 14 and 15.

[Figure]

*Figure R2. Frequency distribution of the slope of $SO_2/NO_2$ from 55 ships.*

6. At least in the Conclusion the authors should discuss not only the advantages but also the limitations of the MAX-DOAS method, which are: - no NO and CO2 measurements (which are the main components inside the plume) - no measurement during twilight and night.

R: Thanks for this suggestion. We have not discussed the limitations of the MAX-DOAS method so much in previous manuscript. Since MAX-DOAS uses solar scattered light as the source, it cannot be measured at night when there is no sunlight, and there is a large error during twilight and rainy observations. Moreover, the spectrometer used in this experiment covers the range of 296~481 nm, while the strong absorption band of NO is 200~230 nm and $CO_2$ has a strong absorption in infrared. For the limitations of MAX-DOAS method in ship emissions monitoring, we have discussed in the Conclusion of the manuscript. Please refer to Line 472-474.

Specific comments:
1. Line 15: It is confusing that the authors mention that the measurements took place in Shanghai and Shenzhen and the next sentence starts with "These three typical measurement sites: : :" In Section 2.2 it becomes clear that the measurements were performed at three sites in two regions –> please clarify in the abstract.

R: "…… in China's ship emission control area (ECA) of Shanghai and Shenzhen, China. These three typical measurement sites are used to……" has been changed to "……in

China's ship emission control area (ECA) of Shanghai and Shenzhen, China. Three typical measurement sites were selected in these two regions to ……" Please refer to Line 14-15.

2. Line 37: consider also CO2 (see first general comment)
R: We have discussed the impact of $CO_2$ emissions from ships in manuscript. Please refer to Line 36.

3. Line 42-43: what kind of important role do the ship emitted pollutants play in air quality, human health and climate? Political regulations, monitoring, enforcement, ⋮ ⋮ ⋮?
R: Pollutants emitted by ships will increase the levels of $NO_2$, $SO_2$ and particulate matter in coastal cities, and the emissions will lead to the decline in urban air quality. Pollution from ships will increase mortality in surrounding areas. Nearly 70% of ship emissions occur within 400 km of coastlines, causing air quality problems through the formation of ground-level ozone, sulphur emissions and particulate matter in coastal areas and harbors with heavy traffic. Besides, Ship emissions have a certain degree of impact on the climate. Studies indicate that the cooling due to altered clouds far outweighs the warming effects from greenhouse gases such as carbon dioxide ($CO_2$) or ozone from shipping, overall causing a negative present-day radiative forcing (RF) (Eyring et al., 2010; Lai et al., 2013; Liu et al., 2016; Yang et al., 2007). Therefore, with the increased awareness of impacts by ship emission, the ship emitted pollutants will be more strictly controlled by Political regulations, monitoring, enforcement and other related departments.

4. Line 53-54: The authors have to distinguish different limits in different ECAs: So the maximum fuel Sulphur Continent (FSC) in ECAs at the US coast and Europe (whole Baltic- and North Sea) is 0.10% S m/m. Inside the Chinese ECA it is 0.50 % S m/m. By 2020 the maximum FSC is 0.50% S m/m all over the world (global Sulphur cap) which is not related to designated ECAs. As the use of exhaust gas treatment systems (Scrubber) is allowed as an alternative, the authors should mention this option, too.
R: China's ECA regulations are different from other regions such as the US and Europe. In China, all ships in the ECAs are required to use fuel with a sulfur content not more than 0.50 % m/m during docking from January 1, 2018. As of January 1, 2019, the ship entering the ECAs should use fuel with a sulfur content of not more than 0.50 % m/m, whether it is sailing or docking. Besides, the maximum FSC in ECAs at the US coast and Europe is 0.10% S m/m. We have distinguished the limits of different ECAs in the manuscript, and also added the content that exhaust gas treatment systems (Scrubber) is allowed as an alternative. Please refer to Line 55-64.

5. Line 57: Are Scrubbers allowed in the Chinese ECA? If yes, this has to men mentioned here, too.
R: Scrubbers is not forbidden in China and we have added this content to the manuscript. Please refer to Line 62.

6. Line 64: "$\because$ usually fast detecting $\because$" –> What is fast (minutes, hours, days, $\because$)?

R: Using the portable rapid analyzer of fuel oil sulfur content can complete the detection of sulfur content of fuel in dozens of minutes, the detection accuracy is controlled within the order of 0.1 ppm. Please refer to Line 70.

7. Line 69: the authors should give a reference (e.g. Kattner et al. 2015, or Seyler et al. 2017)

R: Thanks for the suggestion, we have followed and added these two references in the manuscript. Please refer to Line 74.

8. Figure 2: Do the authors have the copyright for the satellite pictures in Fig. b-d? If not, they have to cite the source.

R: We have marked the source of the picture (b) to (d) were cited from Google Earth. Please refer to Figure 1 in revised manuscript.

9. Figure 3: It looks like graph (b) and (d) are mixed up, because the y-scales do not match the y-scales of graph (a) and (c). Can the authors confirm this? If not, the authors should comment in the corresponding discussion (line 170-175) why the scales are different

R: We have not adjusted the y-scales of graph (a) and (c) in order to make the lines of absorption structure look more obvious. Now we have unified the y-scales of $NO_2$ and $SO_2$ in both cases. Please refer to Figure 2 in the revised manuscript.

10. Figure 3 (line 180-181): Wouldn't it be better co compare measurements with same elevation angle to minimize differences due to different optical length inside the planetary boundary layer (PBL)?

R: According to the specific comments 9 and 10, two spectra measured at the same elevation angle of 5° on June 22, 2018 were selected to re-plot the Figure 3. Please refer to Figure 2, Line 164-172.

[Figure]

*Figure R3. Typical DOAS spectral fitting for SO₂ and NO₂. (a) and (b) show the clean condition of spectrum collected at an elevation angle of 5° at 10:39 LT on 22 June, 2018, while (c) and (d) are the ship plumes polluted case of spectrum measured at an elevation angle of 5° at 09:53 LT on 22 June, 2018. Black lines show the measured atmospheric spectrum and the red line shows the reference absorption cross-section.*

11. Line 190: What is located opposite the berth? Any industry which might emit NOx or SO2?

R: Opposite the berth is the South Channel of the Yangtze Estuary. The only sources of NOx and SO₂ could be the ships emissions on the channel, and the main channel is more than two kilometers away from the berth. Behind the building of Pudong MSB, there are green land and residential area. The container yard was located between the building and berths.

12. Line 196: Why only 10∘ angle for the reference spectrum? At 10∘ the way thru the PBL is still long and therefore might be influenced by emissions. Why the authors did not measure at 90∘ or at least at 65∘ like in Wusong?

R: We can collect reference spectrum at 90° angle in Yantian because the instrument was installed outdoors without obstructions. But the instruments in Waigaoqiao and Wusong are installed indoors. Due to the block of the building, higher elevation angle cannot be achieved. Therefore, we choose a relatively clean orientation as reference spectrum. Please also refer to the responses to Reviewer #2, where we showed the

comparison of DSCDs results using different reference spectrum. The different choice of reference spectrum can affect the absolute value of DSCDs, but not change the characteristics of 2-D spatial distribution of DSCDs.

13. Figure 4: In my opinion this figure is not necessary to explain the measurements. So if the authors want to save some space, they could skip this figure.
R: Thanks to the reviewer's suggestion. Since the observation method of Waigaoqiao is more complicated than the other two places, the Figure 4 can facilitate the description of the experimental scheme more organized and clear. Besides, this figure can help readers better understand the horizontal and vertical observation methods of MAX-DOAS. So we decide to keep it.

14. Line 208-210: please explain with compass direction (e.g. shift to north-west) to compare it to the given wind direction. What dose "MAINLY came from the south" mean? Here only one measurement (a 15 min scan) is discussed. Dose the wind direction changed during this 15 minute scan?
R: Since there is no weather station in Waigaoqiao, we refer to the wind information provided by the Shanghai urban site. Several stations in Shanghai have shown that the wind was from the SSE at 12:00, and the wind direction was also dominated by the southerly wind in the next two hours. So it could be considered that the wind direction has not changed significantly within 15 minutes.

15. Line 220-221: If the authors have "more than a dozen cycles" why they don't show an average of all cycles. The presented cycle is a snapshot and might be not representative to draw conclusions.
R: Please refer to the previous responses to the general comments #3.

16. Line 253: ships in navigation or maneuver? Do the authors can exclude emissions from the industry behind the river (visible in Fig 2 (c)) as a source?
R: The ships were in the state of navigation. According to Figure R4, we can see the main green land areas behind the river, including villages and forest parks. In addition, the opposite of the river is a small dock and a station for transporting containers. Due to the lack of relevant research and observations in this region, it is very difficult to estimate the amount of $NO_2$ and $SO_2$ emitted from the opposite side, and we have to ignore this part of the source compared with the ship emissions on the channel.

[Figure]

*Figure R4. Map of Wusong measurement site and direction of observation, cite from Google Earth.*

17. Line 256-268: Why the authors do only check for the influence of wind speed? I would expect a much bigger correlation with wind direction because the optical length inside the polluted air and therefore the response signal is probably increased when the wind transports the polluted air towards the DOAS. On the contrary the optical length inside the polluted air and therefore the response signal is probably less when the wind transports the polluted air perpendicular to the DOAS (out of the field of view). The authors should check the wind direction dependency for all data. Why constant high wind speed (5-6 m/s is not high) unstable atmospheric conditions? As stated above, in my opinion the wind direction is of high importance as well.

R: Thanks for the suggestion, we did not consider carefully before. We have re-analyzed all the data from January to March 2018 and calculated the hourly mean of the $NO_2$ and $SO_2$ DSCDs for each elevation angle. The wind rose diagrams of $NO_2$ and $SO_2$ at elevation 5° are shown in (a) and (b) of Figure R5. The wind during the observation period mainly comes from NNW. It can be seen that the average of $NO_2$ DSCDs is small under the wind conditions from North. When the wind direction is parallel to the observation direction (i.e. E and W, the viewing direction of the telescope is pointing to the East), the average DSCDs of $NO_2$ is significantly higher than the direction of N. Similarly, the average value of $SO_2$ in the E and W is higher than that in the S and N. It suggests that the optical length inside the polluted air and therefore the response signal is probably increased when the wind transports the polluted air parallel to the DOAS viewing direction.

In order to prove this point more accurately, we have made a statistics in Figure R5 (c) and (d), the perpendicular direction for N and S is the considered wind from 0°±15° and 180°±15°, and the parallel direction for E and W is considered wind from 90°±15° and 270°±15°. It can be seen from (c) and (d) that the $NO_2$ and $SO_2$ DSCDs are quite different in these two types of wind directions. When the wind is parallel to the observation direction (E and W), 34 percent of $NO_2$ is greater than $3.00\times10^{16}$ molec cm$^{-2}$, 31 percent of $SO_2$ is greater than $1.5\times10^{16}$ molec cm$^{-2}$. However, in the perpendicular

direction (N and S), the occurrence of high DSCDs of $NO_2$ and $SO_2$ is significantly less than that of parallel direction. We have added this part to the manuscript. Please refer to Line 289-304.

Besides, because of the average wind speed is less than 3.7 m/s during the observation period, so we set the wind speed higher than 5m/s as the condition of unstable atmospheric.

[Figure]

***Figure R5. The dependence of (a) and (c) of NO₂ and (b) and (d) of SO₂ DSCDs on wind directions from January to March 2018 at elevation 5°.***

18. Line 263-264: The close relation of SO2 and NO2 signal to the flow of ships (better use "ship density) is not clear on March 09 12:00-14:00. At this time the ship density is constant but SO2 and NO2 decrease.

R: As mentioned in manuscript and Figure R5 and R6 in response, the DSCDs of $NO_2$ and $SO_2$ was affected by many factors, including wind speed and direction, as well as ship density. So the DSCDs changes are not exactly the same as the ship density trend. Please also refer to the responses to specific comments 19.

19. Figure 7 and corresponding discussion: If the authors want to correlate SO2 and NO2 with traffic density and meteorological conditions, they should use all data and not

only data from two days. They should create scatter plots (e.g. ship density on x and SO2 on y-axis) to find correlation coefficients. As the authors give a scan time of 7 minutes for the Wusong measurements (Table 1), each box-whisker-plot is based on only 4 measurements. This is not valid for this kind of plots. If the authors use all azimuth angles in one box, this is also not valid, because of different optical length and therefore not comparable measurement conditions.

R: It is true that the amount of data in two days is too little to prove the conclusion of this part. We have added more data during the observation period to show the relationship between ship density and the DSCDs of NO₂ and SO₂ at elevation 5°.

Due to lack of the data of ship density from MSB office, we have to count manually the ship density during each hour based on the real-time photos taken by the instrument. In this way, over fifty photos need to be manually checked and counted for each measured hour, which costs lots of time and also contains uncertainties to some extent. Therefore, we have looked through all the photos from January 30 to February 13, 2018 and another two days in Fig. 7 (January 1 and March 9, 2018). In total, 17 days were used to discuss the relationship between ship density and DSCDs of NO₂ and SO₂, as shown in Figure R6.

In Figure R6, the hollow squares in the middle of the box represent the mean value, and the solid lines in the middle represent the median. The upper and lower edges of the box are 25% and 75% quantiles, respectively. It is found from Figure R6 that as the ship density increases, the hourly mean values of NO₂ and SO₂ show an upward trend. Since the fuel used by the ship is inconsistent, and the speed and direction of win are also affect the DSCDs, it is difficult to find the linear relationship between ship density and DSCDs every hour in this complex environment. However, we have made the linear analysis of the ship density and the corresponding average of the DSCDs (the hollow squares in the middle of the box). The DSCDs of SO₂ has a high correlation coefficient with ship density (R=0.97), while the correlation of NO₂ is relatively weak (R=0.86) due to the more complicated emission sources nearby. We have added this part to the manuscript. Please refer to Line 306-315.

[Figure]

*Figure R6. Relationship between DSCDs of (a) NO₂ and (b) SO₂ and ship density.*

20. Line 276-279: This conclusion might be true, but has to be proved not only by a snapshot but by an average as indicated in the comment above.

R: Please refer to the responses to specific comments 17, 18 and 19.

21. Line 282-283: To support the conclusion that the ships are the main source for NO2, the authors should roughly estimate the influence of the surrounding emission sources (especially the main roads and highway, because this could be a significant NO2 source).

R: Due to the lack of relevant research in the areas of Wusong site, we are unable to obtain the influence of the surrounding emission sources accurately. We have installed two active LP-DOAS (Long-Path DOAS) in the Wusong MSB and Fudan University Jiangwan Campus in March of 2018, respectively. The locations of Wusong and Fudan have been shown in Figure R7 (a), and the red arrow indicates the light path of the two LP-DOAS. The campus is 4 kilometers away from Wusong and is considered to be free of pollution because the campus is almost covered by green spaces and have no major emission sources.

Considering the synchronization of data, concentration of $NO_2$ from March 15 to March 30, 2018 have been analyzed. During the observation, the average value of $NO_2$ in Fudan campus and Wusong site is 12.42 ppb and 30.50 ppb, respectively. In order to make a clearer explanation, we have shown the time series of $NO_2$ in a short segments of three days as an example in Figure R7 (b). We have calculated the difference of $NO_2$ between Wusong and Fudan campus, which is considered to represent the sum of $NO_2$ emissions from ships and surrounding sources at Wusong area. For the red line in Figure R7 (b), the rapidly changing part is discharged by the ships, while the smooth part may come from surrounding emission sources such as roads and highway. After a rough estimation, the proportion of $NO_2$ emitted by ships is more than 47%. However, this estimation are quite rough and more accurate conclusion need be furthered with multiple measurements and technical method.

[Figure]

*Figure R7. The locations of the LP-DOAS measurements at Wusong site and Fudan campus (a), the viewing direction and distance of instrument is indicated by a red arrow, cite from Google Maps, and (b) time series $NO_2$ concentration from March 16 to March 19, 2018.*

22. Line 285-287: The meaning of this sentence in not clear. Do the authors mean that they want to use the complicated MAX-DOAS inland waterway measurements to check ships for compliance with fuel Sulphur regulations? It is not clear how the authors want to compare a theoretical SO2 emission (which unit?) with the MAX-DOAS measurement result (column density). This has to be explained in more detail. In my opinion comparing only SO2 from theoretical emissions to DOAS measurements does not work, because from the DOAS measurements you don't know whether you measure in line or perpendicular to the plume.

R: Thanks for the suggestion. We have not expressed it clearly in this sentence, and we have reorganized the sentence. Please refer to Line 324-329. It is difficult for regulatory authorities to achieve fuel detection for each ship due to the large ship density in complicated inland waterways. So the application of remote sensing technology could provide support to regulatory authorities. The theoretical $NO_2$ an $SO_2$ concentration of plume exhausted from the chimney can be calculated based on the legally sulfur content and ship activity data. Besides, combined with the diffusion model of plume, the theoretical concentration of $SO_2$ on the observation path of MAX-DOAS can be obtained. Therefore, MAX-DOAS can be used to mark the suspicious ships on the complicated inland waterways according to whether the observed $SO_2$ concentration exceeds the theoretical value.

23. Line 292: largest container ports related to what? China, Asia, World?

R: Yantian Port in Shenzhen is the largest single port area with the largest container throughput in China. According to the survey, Shenzhen Port is the third port of the world container port in 2017, and Yantian Port is the main port of Shenzhen Port. So the Yantian Port is one of the largest container port in China and even in the world.

24. Line 305-308: Why the big container vessel does not show any NO2 signal but the tug do so? The container vessel should emit even more NOx than the two tugs. Please discuss.

R: We have not noted this phenomenon in detail before. Due to the unreasonable setting of the color bars, the signal of the big ship looks very weak. We have re-adjusted the color bars in Figure R8 to make the emission signals of several ships look more intuitive. According to Figure R8 (b), we can observe that the signal of $NO_2$ emissions from large ships are obvious. Please also refer to Figure 10 in manuscript.

[Figure]

*Figure R8. Measured DSCDs of (a) SO₂ and (b) NO₂ during 12:55~14:20 and live photos taken by the camera at (c) 12:56 and (d) 13:22 on May 26, 2018.*

25. Line 314-316: This sentence put a question mark onto the measured plumes at 13:00 and 13:30. So the authors have to state that for the two earlier measurements no other ship could have caused the high signals (e.g. based on AIS analysis).

R: We have determined through AIS information and on-site records that there are no other ship emissions disturbances for the two earlier measurements, and we have made corresponding corrections to this sentence. Please refer to Line 358-359.

26. Line 323: How the emissions are related to the operational status (was not discussed before)?

R: In general, under given conditions, emissions from vessels for propulsion engines and auxiliary engines can be estimated by equations (1) - (3) (Zhang et al., 2017).

$$E = Load \times Activity \times EF \times FCF \times CF \qquad (1)$$
$$Load = MCR \times LF \qquad (2)$$
$$LF = (V_{actual} / V_{maximum})^3 \qquad (3)$$

Where
E = emissions, g;
Load = engine power, kW;
MCR = maximum continuous rating, kW;
LF = load factor, dimensionless;

$V_{actual}$ = actual speed, knots;

$V_{maximum}$ = maximum speed, knots;

Activity = ship activity time, h;

EF = emission factor, g/kWh;

FCF = fuel correction factor, dimensionless;

CF = control factors for emission reduction measures, dimensionless.

According to formula (1), when the EF, FCF, and CF are constant, the ship's emissions are closely related to the power, activity time and the speed of ship. Unfortunately, we cannot get all the parameters in the formula in this study. However, it can be found by estimation that the emissions will increase when the power increases. We have added this part of the discussion to the manuscript. Please refer to Line 407-408.

27. Line 326-327: "we try to further detailed SO2 emissions from the measurement" –>the meaning is not clear. Do the authors want to say that they try to analyze detected plumes in more detail?

R: Thanks for the suggestion, we did not express this sentence clearly. We try to analyze the detected plumes in more detail. This sentence has been re-phrased, please refer to Line 369-370.

28. Line 339: Which mathematical method? How the baseline is calculated (e.g. running mean or median; which averaging interval?)?

R: The mathematical algorithm used here is BESDS (baseline estimation and denoising using sparsity). Specifically, the baseline is modeled as a low-pass signal and the series of peaks is modeled as sparse with sparse derivatives. Moreover, to account for the positivity of peaks, both asymmetric and symmetric penalty functions are utilized (Ning et al., 2014). The specific methods and principles we have supplemented in the manuscript. Please refer to Line 238-240, and also refer to the responses to Reviewer #2.

29. Line 351-355: The use of fuels with different Sulphur content is one possible explanation for the observation of different SO2/NO2 ratios. The age of the plume and the direct NO2 emissions is another plausible explanation. As already mentioned in the first general comment, ships mainly emit NO but not NO2. Therefore, directly at the stack the SO2-NO2 ratio is highest. When the plume ages NO2 is formed and the SO2-NO2 ratio decreases. NO2 emissions are dependent on the kind of engine and the burning temperature of the engine as well. The authors have to consider this in their discussion.

R: Thanks for the suggestion, we have considered the impact of NO and added this part of the discussion to the manuscript. Please refer to the previous responses to the general comments #1 and #2.

30. Line 359-361: Is the main engine really operated with fuel with higher Sulphur content than the auxiliary engine? Do the authors have a source for this statement? I

thought inside the Chinese ECA the maximum allowed fuel Sulphur content is equal to that allowed at berth.

R: The detailed questionnaire of vessel information was obtained by boarding inspection. The inspector took fuel samples of several ships and brought them back to the laboratory for testing. The results shows that the sulfur content data in the questionnaire can be considered to be accurate after verification. The Figure R9 provides examples of two ships. Besides, there is no 0.5% requirement for ships in navigation during our observation period in 2018. Please also refer to the responses to the specific comments #4 and #31.

[Figure]

*Figure R9. The detailed questionnaire of vessel information: two example ships.*

31. Line 381: Do the authors mean vessel #IX instead of cargo #IV? What about vessel #V and #VIII? Are they allowed to use fuels with Sulphur content above 3% inside the Chinese ECA? I thought the limit is 0.5%.

R: The "#IV" has been changed to "#IX". Please refer to Line 431.

In China, all ships in the ECAs are required to use fuel with a sulfur content not more than 0.50 % m/m during docking from January 1, 2018. As of January 1, 2019, the ship entering the ECAs should use fuel with a sulfur content of not more than 0.50 % m/m, whether it is sailing or docking. So, there is no 0.5% requirement for ships in navigation during the observation period in 2018.

32. Line 391-394: from the SO2-NO2 ratio displayed in Fig. 12 it is not obvious which is the "irregular observed ratio". I do agree that if the SO2-NO2 ratio is above 1.5 this is an indication for the use of fuel with high Sulphur content. But from this point of view vessel III, V, VIII, and IX should be indicated as non-compliant. Is it possible to estimate a kind of detection limit for the observation of non-compliant vessels? Is it

possible to distinguish between 0.4 (compliant) and 0.8 (non-compliant)? This would address also the conclusion (Line 410-412).

R: Thanks to the reviewers, vessel III, V, VIII, and IX were considered as ships that use unqualified fuel due to the high ratio of $SO_2/NO_2$. According to the nine samples in Figure 12, the detection limit for the observation of non-compliant vessels we set is 1.5. More legally fuel sulfur content and ship activity data are needed to help us find a reasonable way to distinguish the compliant and non-compliant vessels. It is difficult to distinguish between 0.4 (compliant) and 0.8 (non-compliant) until now, which can be more accurately estimated along with the data of load factor and emission factor during the actual operation in the future. Some explanations have been added to the manuscript. Please refer to Line 440-447.

Technical corrections
Line 15: ships instead of ship
R: The "ship" has been changed to "ships". Please refer to Line 16.

Line 22: ad "the" in front of SO2/NO2: : :
R: The "the" have been added. Please refer to Line 23.

Line 26: Combining instead of Combined
R: The "Combined" has been corrected to "Combining". Please refer to Line 27.

Line 27: What is meant with "logical Sulphur content"
R: By combining the measured data with the actual operating parameters of the ship, the ship's emission model and the diffusion model of the plume, the sulfur content of the fuel used by ship will be calculated. Here we used "logical sulphur content" to represent the assumed S% in emission model estimation, which should be legal.

Line 28: "more accurate way" –> than what?
R: Here we describe the prospective of MAX-DOAS application for the surveillance of ship emissions. In this study, the empirical ratio of $SO_2/NO_2$ was only concluded based on the DOAS measurements and several samples of ships. By combining with ship emission estimated by actual operation parameters and logical sulfur content, more accurate ratio of $SO_2/NO_2$ for compliance could be obtained, which can improve the accuracy of the surveillance of ship emissions by MAX-DOAS measurements.

Line 44: ad "of" in front of "kilometers"; remove "the" in front of "most.
R: We have corrected them. Please refer to Line 45-46.

Line 53: areas instead of zones
R: The "zones" has been changed to "areas". Please refer to Line 55.

Line 57: should or must?
R: The meaning expressed here is "should".

Line 85: close instead of closed
R: The "closed" has been changed to "close". Please refer to Line 90.

Line 95: ad "the" in front of instrument
R: We have added "the". Please refer to Line 101.

Line 105-106: "*: : :*and stored in form of spectrum" –> It is not clear what is meant. The MAX DOAS instrument records spectra which represents the intensity of scattered sunlight at different wave length (please give the scan interval). For each viewing direction and measurement interval a separate spectrum is recorded.
R: The spectrometer records the intensity of solar scattered light in the wavelength range from 296 nm to 481 nm, there are 1024 data points with an average scan interval of 0.18 nm. The spectrum is stored as a file, it not only contains the light intensity at 1024 bands, but also contains the corresponding time, date, solar zenith angle, solar azimuth angle, and elevation angle, etc. Please refer to Line 143.

Table 1: The measurement sites name and locations should start in the same line the operations AZ starts. At the moment it is little bit confusing that the line where the AZ is given starts above the site names.
R: We have followed the suggestion and made the correction. Please refer to Table 1.

Table 1: The authors should give the number of scan cycles, as well. As the measurements in Waigaoqiao last less than one day and one can takes 15 minutes I think there are not many scans available for analysis.
R: Please refer to the previous responses to the general comments #3.

Line 186: ad "of" in front of "more than"
R: The "of" have been added in front of "more than". Please refer to Line 199.

Line 189: ad "of" in front of "ships at"
R: The "of" have been added in front of "ships at". Please refer to Line 202.

Line 194: remove "can" in front of "covers about"
R: The "can" has been removed. Please refer to Line 207.

Line 204: increase instead of increases
R: The "increases" has been changed to "increase". Please refer to Line 217.

Line 242: write "Beside of ocean-going ships, inland waterway vessels also contribute significantly to the amount of ship emissions*: : :* " instead of "Besides oceangoing ship emissions, inland waterway vessels also contributed significantly to the ship emissions*: : :*" –> not the emissions, but the ships are ocean-going :-)
R: This sentence has been modified in the manuscript. Please refer to Line 255.

Line 245: close instead of closed

R: The "closed" has been changed to "close". Please refer to Line 258.

Line 246-247: This sentence is not clear. I suggest to split it.

R: We have split this sentence. "It is only channel to the upstream of Huangpu River. There are some non-container terminals near the measurement site, which mainly handles goods in domestic trading". Please refer to Line 259-260.

Line 251: ad a space (direction of)

R: The space has been added. Please refer to Line 264.

Line 252: lane instead of lanes

R: The "lanes" has been changed to "lane". Please refer to Line 265.

Line 256: use singular instead of plural: impact of ship traffic; measurement data

R: We have corrected them. Please refer to Line 270.

Line 273: what do the bars and the stars do represent?

R: The bars is called whisker line, whiskers extend from each end of the box to the internal and external limits. The star is composed of "-" and "×", "-" represents the maximum and minimum, and "×" are 1% and 99% quantiles. Please refer to Line 285-287.

Line 291: see instead of See

R: We have corrected it. Please refer to Line 333.

Line 299: remove "orderly" in front of "inbound"

R: The "orderly" has been omitted. Please refer to Line 340.

Line 300: 2018 instead of 2019; two dots at the end

R: We have corrected them. Please refer to Line 342.

Line 302: remove "were" in front of "occurred"

R: The "were" has been removed. Please refer to Line 344.

Line 310: was instead of were

R: The "were" has been changed to "was". Please refer to Line 352.

Line 323: "more or less" –>please be more precise!

R: The "more or less" has been removed. Please refer to Line 366.

Line 325: "for" instead of "with the"

R: The "with the" has been changed to "for". Please refer to Line 368.

Line 330: "was" instead of "can be"
R: The "can be" has been changed to "was". Please refer to Line 373.

Line 331: "stand for the peak concentration" –>do the authors mean "represent the peak concentration"?
R: The "stand for" has been changed to "represent". Please refer to Line 373.

Line 337-338: It is difficult to get the meaning of this sentence. Do the authors want to say that with temporal high resolved measurements (60 sec) it was possible to resolve individually the plume signals of passing ships? Please rephrase.
R: Here we want to express that only a single elevation angle is observed instead of scanning all elevation angles can help us get more data at elevation 7°. Please refer to Line 378-379.

Line 340: Comma after DSCDs
R: The "comma" has been added after "DSCDs". Please refer to Line 383.

Line 342: "are present" behind cm-2
R: The "are present" has been added. Please refer to Line 385.

Line 345: remove "In addition" in front of "the increases of pollutants"
R: The "In addition" has been removed. Please refer to Line 388.

Line 368: "on" instead of "about"
R: The "about" has been changed to "on". Please refer to Line 419.

Line 374: "is" instead of "are"
R: The "are" has been changed to "is". Please refer to Line 425.

Line 379: ad "the" in front of "plume"
R: The "the" has been added in front of "plume". Please refer to Line 429.

Line 380: "increase" instead of "growth"
R: The "growth" has been changed to "increase". Please refer to Line 429.

Line 381: "to note" instead of "noted"; "circle instead of "dot"
R: We have corrected them. Please refer to Line 430.

Line 390: ad "high" in front of "sulfur content"
R: The "high" has been added in front of "sulfur content". Please refer to Line 438.

Line 393: "compliance monitoring" instead of "compliance"
R: The "compliance" has been changed to "compliance monitoring" Please refer to Line

441.

Line 397: write "In this study we performed MAX-DOAS measurements to observe ship emissions of $SO_2$ and $NO_2$ in Shanghai*∶∶*" instead of "In this study, we have performed the MAX-DOAS measurements observe the ship emissions of $SO_2$ and $NO_2$ in Shanghai*∶∶*"

R: We have improved it. Please refer to Line 452.

Line 400: delete comma

R: The comma has been deleted. Please refer to Line 455.

Line 402: better "*∶∶∶* are correlated to ship traffic density at stable and unstable atmospheric*∶∶*"

R: We have improve it. Please refer to Line 457-458.

**References**

Eyring, V., Isaksen, I. S., Berntsen, T., Collins, W. J., Corbett, J. J., Endresen, O., Grainger, R. G., Moldanova, J., Schlager, H., and Stevenson, D. S.: Transport impacts on atmosphere and climate: Shipping, Atmos. Environ., 44, 4735–4771, https://doi.org/10.1016/j.atmosenv.2009.04.059, 2010.

Han, S., Bian, H., Feng, Y., Liu, A., Li, X., Zeng, F., Zhang, X.: Analysis of the Relationship between $O_3$, NO and $NO_2$ in Tianjin, China., Aerosol. Air. Qual. Res., 11: 128–139, https://doi.org/10.4209/aaqr.2010.07.0055, 2011.

Jalkanen, J. P., Johansson, L., Kukkonen, J.: A comprehensive inventory of ship traffic exhaust emissions in the European sea areas in 2011., Atmos. Chem. Phys., 16, 71–84, https://doi.org/10.5194/acp-16-71-2016, 2016.

Lai, H. K., Tsang, H., Chau, J., Lee, C. H., McGhee, S. M., Hedley, A. J., and Wong, C. M.: Health impact assessment of marine emissions in Pearl River Delta region, Marine Pollution Bulletin, 66, 158-163, https://doi.org/10.1016/j.marpolbul.2012.09.029, 2013.

Liu, H., Fu, M., Jin, X., Shang, Y., Shindell, D., Faluvegi, G., Shindell, C., and He, K.: Health and climate impacts of ocean-going vessels in East Asia, Nat. Clim. Change., 6, 1037-1041, 2016.

Mazzeo, N. A., Venegas, L. E., Choren, H.: Analysis of NO, $NO_2$, $O_3$ and NOx concentrations measured at a green area of Buenos Aires City during wintertime. Atmos. Environ., 39: 3055-3068, https://doi.org/10.1016/j.atmosenv.2005.01.029, 2005.

Mellqvist, J.; Beecken, J.; Conde, V.; Ekholm, J.: Surveillance of Sulfur Emissions from Ships in Danish Waters. Report to the Danish Environmental Protecion Agency. Available online: http://dx.doi.org/10.17196/DEPA.001, 2017.

Ning, X., Selesnick, I., and Duval, L.: Chromatogram baseline estimation and denoising using sparsity (BEADS). Chemom. Intell. Lab. Syst., 139, 156-167, https://doi.org/10.1016/j.chemolab.2014.09.014, 2014.

Singh, H. B.: Reactive nitrogen in the troposphere, Environ. Sci. Technol., 21, 320–327, https://doi.org/10.1021/es00158a001, 1987.

Song, C. H., Chen, G., Hanna, S. R., Crawford, J, Davis, D. D.: Dispersion and chemical evolution of ship plumes in the marine boundary layer: Investigation of $O_3$/NOy/HOx chemistry. J. Geophys. Res: Atmos., 108(D4), https://doi.org/10.1029/2002JD002216, 2003.

Yang, D., Kwan, S., Lu, T., Fu, Q., Cheng, J., Streets, D. G., Wu, Y., and Li, J.: An emission inventory of marine vessels in shanghai in 2003. Environ. Sci. Technol., 41(15), 5183-5190, https://doi.org/10.1021/es061979c, 2007.

Zhang, Y., Gu, J., Wang, W., Peng, Y., Wu, X., and Feng, X.: Inland port vessel emissions inventory based on Ship Traffic Emission Assessment Model–Automatic Identification System, Adv. Mech. Eng., 9(7), 1–9, https://doi.org/10.1177/1687814017712878, 2017.

---

## Author Comment (AC2) · 25 Jul 2019

**Response to comments from reviewer #2**

We thank the reviewers for the constructive comments and suggestions, which are very positive to improve scientific content of the manuscript. We have revised the manuscript appropriately and addressed all the reviewers' comments point-by-point for consideration as below. The remarks from the reviewers are shown in black, and our responses are shown in blue color. All the page and line numbers mentioned following are refer to the revised manuscript without change tracked.

Reviewer

The paper presented by Cheng et al. has reported the shore-based MAX-DOAS measurements of ship emitted SO2 and NO2 under three different conditions in China's ship emission control area (ECA), i.e. ship docked at berth, navigation in the inland waterway and inbound/outbound in the deep water port. Although the detection of SO2 and NO2 by MAX-DOAS has been developed for many years, the employments for ship emission surveillance are an interesting application of the MAX-DOAS technique. I think the manuscript fits to the scope of ACP, especially for this special issue. I recommend publication after the authors addressed the following comments.

Major concerns:

1. The authors use the SO2 and NO2 DSCDs measured at different elevations for the evaluation of ship emissions. However, the vertical distribution of background SO2 and NO2 are quite different. It is not clear that how do the authors separate the ship emissions of SO2 and NO2 signal from the background? This information has to be supplemented in section 2.

R: The explanation about the difference of $SO_2$ and $NO_2$ signal of ship emissions and background has not been discussed in detail before. Now we have added it in Section 2.3. Please refer to Line 185-195. In order to better demonstrate the $NO_2$ and $SO_2$ concentration in background and emission signal, several typical cycles in June 29th were selected as examples, the selected cycles was boxed out in Figure R1. The data marked with the red and gray shadow is the DSCDs of signal and background, and these two cases have been further shown in Figure R2.

[Figure]

*Figure. R1. Diurnal variations of DSCDs of (a) NO₂ and (b) SO₂ on 29 June 2018.*

Figure. R2 shows the vertical distributions of NO₂ and SO₂ DSCDs with the elevation angle when there is a ship passing through and not. It can be observed that the DSCDs of NO₂ and SO₂ decrease slowly with increasing angle under clean conditions, during which the maximum values of NO₂ and SO₂ DSCDs are $5.03\times10^{16}$ molec cm$^{-2}$ at elevation 3° and $1.78\times10^{16}$ molec cm$^{-2}$ at elevation 2°, respectively.

[Figure]

*Figure. R2. The distributions of (a) NO₂ and (b) SO₂ DSCDs with elevation angle in ship emission signal and background on June 29, 2018.*

In contrast, the NO₂ and SO₂ DSCDs increased significantly when ships passed, showing the maximum values of NO₂ and SO₂ DSCDs of $7.36\times10^{16}$ molec cm$^{-2}$ at elevation 5° and $4.15\times10^{16}$ molec cm$^{-2}$ at elevation 5°, respectively. And the highest value of SO₂ generally appears between elevation angle 5° and 10°. Therefore, it can

be concluded that the signal of ship emissions of $SO_2$ and $NO_2$ can be easily identified and separated from the background clean conditions when there is a ship passing nearby, which can be further confirmed by the AIS information, on-site photos and records, etc.

2. The sectioning of section 2 is not very logical. I suggest the authors follow the order of "instrument", "spectral retrieval" and "ship emissions identification".
R: Thanks for the constructive suggestion. We have followed the order of "instrument", "spectral retrieval" and "ship emissions identification", and reorganized the Section 2. Please refer to Section 2 from Line 109-195.

3. In section 2.3, the SO2 and NO2 DSCDs are retrieved at different spectral ranges. How do the authors compensate the effect of wavelength dependency? If it is not considered in the retrieval, an error analysis is required.
R: The configuration of $SO_2$ and $NO_2$ spectral analysis was based on many previous studies, e.g. Hendrick et al., 2014; Irie et al., 2011; Seyler et al., 2017; Wang et al., 2014. So the common fitting window of 307.5-320 nm and 338-370 were used for $SO_2$ and $NO_2$, respectively. As it can be seen in Fig. R3, the strong absorption band of $SO_2$ is below 325 nm, where the $NO_2$ absorption are relatively weak. It means that the wavelength band of $SO_2$ analysis window should be shorter than that of $NO_2$.

[Figure]

***Figure R3. Absorption cross section of NO₂ and SO₂ in the wavelength range of 300~400 nm.***

Since it is obvious that the $SO_2$ analysis cannot be performed well in longer wavelength over 325 nm, we have tried the analysis of $NO_2$ with the same fitting interval of $SO_2$ in 307.5~320 nm. As shown in the Fig. R4 (a), we found that the $NO_2$ DSCD values from fitting window of 307.5~320 nm are larger than that in 338-370 nm and simultaneously shows considerable uncertainties. In addition, Fig. R4 (b) and (c) show that fitting interval of 307.5~320 nm for $NO_2$ generates even larger RMS and DSCDs error compared to the results from fitting within 338~370 nm. It suggests that the DSCDs from same fitting window will bring large uncertainty and error in the results. Finally, we decided to use the different fitting intervals for $SO_2$ and $NO_2$.

[Figure]

*Figure R4. Comparison of NO₂ retrieval with different fitting intervals of 307.5-320 nm and 338-370 nm on 26 June 2018: (a) NO₂ DSCD with error bars, (b) RMS and (c) DSCD error.*

Regarding to the compensation of wavelength dependency effect, we think the way to use the ratio of $SO_2$ to $NO_2$ DSCDs to identify the ship emission will not be impacted by the effect of wavelength dependency. Because the fixed analysis fitting window was applied for all campaigns and the ratio will not contain the wavelength dependency effect (or in presence as the systematic deviations).

4. Sect. 3.1, In the 2D scanning, the authors used the reference spectrum measured at azimuth angle of 10, however, it can be seen from Fig. 2(b) that this direction are still pointing to the berth. How to confirm the impacts of ship emission in the reference spectrum has been excluded? Alternatively, how to evaluate the uncertainties on the absolute value of DSCDs due to this?

R: We agreed with this point. In Section 3.1, it aims to prove that MAX-DOAS can recognize the spatial distribution of emission plume. Due to the limitation of the instrumental installation, the zenith-sky spectrum cannot be collected and used for the reference spectrum. So we have to select the measured spectrum at a relatively clean horizontal angle as the reference spectrum, i.e. elevation 7° at azimuth 10°. The 2-D distribution of retrieved $NO_2$ and $SO_2$ DSCDs were displayed in Fig. R5 (a) and (b). Under the same fitting configuration, the measured spectrum collected at elevation 7° at azimuth 30° in the 2-dimensional scanning cycle was also selected as reference spectrum for analysis, and the distribution of $NO_2$ and $SO_2$ DSCDs were shown in Fig.

R5 (c) and (d).

Fig. R5 (e) shows the difference between DSCD of NO₂ obtained by two analysis configurations. The difference between Fig. R5 (a) and Fig. R5 (c) were averaged at is $1.23 \times 10^{16}$ molec cm⁻², and no obvious difference in spatial distribution. This result indicates that the selection of reference spectrum may affect the absolute value, however, do not change the 2-D distribution of retrieved NO₂ DSCDs. Similarly, Fig. R5 (f) shows the difference in SO₂ between Fig. R5 (b) and Fig. R5 (d), and the average value of Fig. R5 (f) is $4.14 \times 10^{15}$ molec cm⁻². Therefore, we choose the spectrum with less trace gas absorption as the reference according to Fig. R5.

[Figure]

*Figure R5. 2-D distributions of measured DSCDs of (a) NO₂ and (b) SO₂ using a reference spectrum collected at elevation 7° and azimuth angle of 10°; and DSCDs of (c) NO₂ and (d) SO₂ using spectrum measured at elevation 7° and 30° azimuth as the reference, (e) and (f) is the difference values between (a) and (c), (b) and (d).*

5. Both in Sect. 3.1 and 3.3, the authors used the mathematic method to the slowly

change of DSCDs in temporal pattern. I think the author should introduce something more about why this method can be used here? And the basic principle? In line 340, how to prove that the baseline represents the diurnal variations of DSCDs mostly due to the change of light path caused by solar zenith angle and the background emissions? R: The mathematical algorithm used here is BESDS (baseline estimation and denoising using sparsity). Specifically, the baseline is modeled as a low-pass signal and the series of peaks is modeled as sparse with sparse derivatives. Moreover, to account for the positivity of peaks, both asymmetric and symmetric penalty functions are utilized. Figure. R6 (a) shows the original data before processing, (b) shows the peak after removal of the baseline, while the black line in (c) represents the baseline and (d) is the residual. More details can be referred to Ning et al., 2014. The specific methods and principles we have supplemented in the manuscript. Please refer to Line 238-240.

[Figure]

*Figure R6. Processing of noisy chromatogram data using BEADS. (a) Chromatogram data with additive noise. (b) Estimated peaks. (c) Estimated baseline. (d) Residual. (Cited from Ning et al., 2014)*

Affected by the solar zenith angle, the light path decreased initially, followed by an increase during the day, which is consistent with the trend presented by the baseline of DSCDs in Figure 12. Besides, Figure R7 shows the comparison between baseline and data of Yantian monitoring station for six days during the June 2018. The comparison

of hourly mean $SO_2/NO_2$ of baseline and the ground-surface in-situ measurement at Yantian shows that these two datasets agreed well with each other with a correlation coefficient R of 0.82, suggesting that the information of $SO_2/NO_2$ in the baseline are quite consistent with the that of the ambient.

[Figure]

*Figure R7. (a) The comparison of hourly mean $SO_2/NO_2$ of baseline and the ground-surface in-situ measurement at Yantian, and (b) the relationship of $SO_2/NO_2$ between the MAX-DOAS and Yantian.*

6. The authors have mentioned that it is difficult to distinguish the single ship plume. How do the authors derive the emissions from different vessels (Figure 11)? How the data are filtered? What is the error?

R: Thanks for the suggestion. Due to the large density of ships and the wide variety of ships at the measurement site of Wusong, we have mentioned in Section 3.2 that it is difficult to distinguish the single ship plume in the busy inland waterway. However, for the observation site in Yantian, Shenzhen, the atmospheric background is cleaner, and the density of the vessels is much less than that of Wusong site. We are able to distinguish the single ship plume based on changes in DSCD of $SO_2$ and $NO_2$ in Section 3.3. The increment of DSCDs can be considered as the consequence of ship emission. Besides, we also verify the operation of the ship based on information such as on-site records and AIS. Please also refer to the previous responses to the comment 1 of the major concerns.

Minor comments

1. What is the typical error of the measurements? Please put the error bars on figure 6, 10 and 11.

R: Please refer to Figure 6, Figure 12 and 13 in manuscript. We have also showed them here as Fig. R8, R9 and R10.

[Figure]

*Figure R8. Time series of DSCD of (a) NO₂ and (b) SO₂ measured at 4° elevation angle in three azimuths on August 28, 2017.*

[Figure]

*Figure R9. Diurnal variations of DSCDs of (a) NO₂ and (b) SO₂ measured at 7° elevation angle on 26 June 2018.*

[Figure]

***Figure R10. The relationship between SO₂ and NO₂ emitted by several typical vessels, the letter "O" indicates the outbound vessels, "I" indicates the inbound vessels, and "T" indicates the tugboat.***

2. Figure 11 is very busy. It is difficult to see the differences between each species. Maybe the authors can separate it into 2 to 3 subplots. More detailed caption is required.
R: Thanks for the suggestion. The previous Figure 11 was divided into two subplots and added error bars. Please refer to (a) and (b) of Figure 13 in manuscript and Figure R10 above. In addition, the (c) and (d) of Figure 13 show the relationship between SO₂ and NO₂ emitted by other 15 typical vessels during the observation period, as suggested by Reviewer #1.

Technical corrections
Line 17, "berth" to "berths"
R: The "berth" has been corrected to "berths". Please refer to Line 18.

Line 105, "instruments" to "instrument", "observe" to "observes"
R: The "instruments" has been changed to "instrument", the "observe" has also

corrected to "observes". Please refer to Line 142.

Line 111, "less trace gas absorptions"
R: The "small" has been changed to "less". Please refer to Line 146.

Line 112, what is the slope column concentration? It should be the slant column density.
R: We have corrected it to "slant". Please refer to Line 147.

Line 122, "impacted by"
R: The "impacted" has been changed to "impacted by". Please refer to Line 183.

Line 174, "unqualified NO2 and SO2 DSCDs" to "unsatisfied spectral fitting", and "fitting results" to "DSCDs results".
R: The "unqualified $NO_2$ and $SO_2$ DSCDs" has been changed to "unsatisfied spectral fitting", the "fitting results" has been changed to "DSCDs results". Please refer to Line 170.

Table 1 title, "operative" to "operation"; in the line of "Yantian", "Smaller" with unnecessary capital letter.
R: The "operative" has been changed to "operation", the unnecessary captical letter of "Smaller" has been corrected. Please refer to Table 1, Line 131.

Table 2, whether the O4 absorption was included in the SO2 fitting range? What's meaning of symbol "–" standing for here?
R: The $O_4$ absorption was not included in the $SO_2$ fitting range, and the "--" was changed to "/". Please refer to Table 2.

Line 191, "multiple berth" to "multiple berths"
R: The "multiple berth" has been changed to "multiple berths". Please refer to Line 204.

Line 226, "the residual after background subtraction"
R: The "the residual of background subtraction" has been changed to "the residual after background subtraction". Please refer to Line 238.

Line 247, "boxes serving"?
R: The "boxes serving" has been changed to "goods". Please refer to Line 260.

Line 268, "around 5 mâA˘ cs-1 on March 9" ´
R: We have added "on" before the date of "March 9". Please refer to Line 281.

Line 277, "impacting" to "influencing"
R: The "impacting" has been changed to "influencing". Please refer to Line 318.

Line 292 and 293, "meters" can be shorten as "m"

R: We have corrected it. Please refer to Line 334 and 335.

Line 300, there are two dots in the end of the sentence. Please delete one.
R: The excess dot has been deleted. Please refer to Line 342.

Fig. 11, I suggest to also indicate the inbound and outbound status of the vessels to easily exam the relationship with slope.
R: Figure 11 in the manuscript has been modified. We have added the letter "O" to indicate the outbound vessels, "I" for the inbound vessels, and "T" for the tugboat. Please refer to the Figure 13, Line 415.

Line 371, SO2-to-NO2 > SO2/NO2, also in the rest of the manuscript.
R: The "$SO_2$-to-$NO_2$" in the manuscript has been changed to "$SO_2/NO_2$". Please refer to Line 424 and other places.

Line 381, IV or IX?
R: It should be IX. Please refer to Line 431.

Line 405, where is the 2-D DSCDs map at Yantian in manuscript?
R: It was a mistake. Since the experiment at Yantian only observes a single azimuth, there is no 2-D DSCDs map. We have corrected it. Please refer to Line 459.

Line 390 and 410, what the ratios of SO2/NO2 of inbound vessels and tugboat? Lower than 1.3 or 1.5? Please keep the consistency of description.
R: We have kept them consistent and the value is determined to be 1.5. Please refer to Line 437 and 465.

**References**

Hendrick, F., Müller, J.-F., Clémer, K., Wang, P., De Mazière, M., Fayt, C., Gielen, C., Hermans, C., Ma, J. Z., Pinardi, G., Stavrakou, T., Vlemmix, T., and Van Roozendael, M.: Four years of ground-based MAX-DOAS observations of HONO and $NO_2$ in the Beijing area, Atmos. Chem. Phys., 14, 765-781, https://doi.org/10.5194/acp-14-765-2014, 2014.

Irie, H., Takashima, H., Kanaya, Y., Boersma, K. F., Gast, L., Wittrock, F., Brunner, D., Zhou, Y., and Van Roozendael, M.: Eight-component retrievals from ground-based MAX-DOAS observations, Atmos. Meas. Tech., 4, 1027–1044, https://doi.org/0.5194/amt-4-1027-2011, 2011

Ning, X., Selesnick, I., and Duval, L.: Chromatogram baseline estimation and denoising using sparsity (BEADS). Chemom. Intell. Lab. Syst., 139, 156-167, https://doi.org/10.1016/j.chemolab.2014.09.014, 2014.

Seyler, A., Wittrock, F., Kattner, L., Mathieu-Üffing, B., Peters, E., Richter, A., Schmolke, S., and Burrows, J. P.: Monitoring shipping emissions in the German Bight using MAX-DOAS measurements. Atmos. Chem. Phys., 17, 10997–11023,

https://doi.org/10.5194/acp-17-10997-2017, 2017.

Wang, T., Hendrick, F., Wang, P., Tang, G., Clémer, K., Yu, H., Fayt, C., Hermans, C., Gielen, C., Müller, J.-F., Pinardi, G., Theys, N., Brenot, H., and M. Van Roozendael.: Evaluation of tropospheric $SO_2$ retrieved from MAX-DOAS measurements in Xianghe, China., Atmos. Chem. Phys., 14, 11149–11164, https://doi.org/10.5194/acp-14-11149-2014 , 2014.

---

## Referee Report (RR1)

Referee comments on the revised manuscript "Surveillance of SO2 and NO2 from ship emissions by MAX-DOAS measurements and implication to compliance of fuel sulfur content" of Cheng et al., 2019 for submission in ACP

The authors addressed all comments and suggestions in an adequate way. The corrections significantly improve the quality of the revised manuscript. However, the authors reply to some of the reviewer's comments in detail only in the response to comments file, but did not update the manuscript. As the explanations in the reply are of good quality which does improve the understanding of the manuscript, I do recommend considering the manuscript for publication in ACP with the below minor changes:

I do reply to selected authors reply on the first reviewer #1 and #2 comments. The initial reviewer's comments are indicated in black, the authors reply in blue and my comments to the authors reply in green.

Reviewer #1

3. Results presented in section 3.1 do represent only one measurement at one day. If the authors have "more than a dozen cycles" (line 220-221), why they don't show an average of all cycles. The presented cycle is a snapshot and might be not representative to draw conclusions. Well, even an average over one afternoon does represent only a snapshot of the situation. The authors should consider this in their discussion or should skip this section.
R: There may be some misunderstandings due to the unclear description. The purpose of Section 3.1 is to investigate the 2-D distribution of retrieved $NO_2$ and $SO_2$ DSCDs using 2-D scanning measurement of MAX-DOAS. The hotspots of $NO_2$ and $SO_2$ are related to the azimuth of the berth where the ship is docked and the corresponding ship operation status. It is obviously that the 2-D distribution of $NO_2$ and $SO_2$ DSCDs changes with time, so that it seems unreasonable to show the average of all scan cycles, which may weaken the spatial distribution of hotspots. However, the results of the rest of the cycles were also shown in Figure 6. It can be seen that the concentration at a given azimuth and elevation is constantly changing throughout the day.
→ The authors could summarize this description and add it to the manuscript introduction

11. Line 190: What is located opposite the berth? Any industry which might emit NOx or SO2?
R: Opposite the berth is the South Channel of the Yangtze Estuary. The only sources of NOx and $SO_2$ could be the ships emissions on the channel, and the main channel is more than two kilometers away from the berth. Behind the building of Pudong MSB, there are green land and residential area. The container yard was located between the building and berths.
→ Ok, but the authors should add a short statement to the manuscript that there are no other sources for SO2 and NO2 than emissions from ships.

16. Line 253: ships in navigation or maneuver? Do the authors can exclude emissions from the industry behind the river (visible in Fig 2 (c)) as a source?
R: The ships were in the state of navigation. According to Figure R4, we can see the main green land areas behind the river, including villages and forest parks. In addition, the opposite of the river is a small dock and a station for transporting containers. Due to the lack of relevant research and observations in this region, it is very difficult to estimate the amount of $NO_2$ and $SO_2$ emitted from the opposite side, and we have to ignore this part of the source compared with the ship emissions on the channel.
→ Ok, but if the authors cannot exclude this influence the measurements, they have to make a short note in the manuscript. E.g. "A small dock and a station for transport container are

located opposite the river, which might slightly influence the measurements." or something similar.

*Figure R6. Relationship between DSCDs of (a) NO₂ and (b) SO₂ and ship density.*
Corresponding to Fig. 9 in the revised manuscript
→ In the figure caption the authors have to explain the meaning of the squares, stars and bars or have to refer to Figure 7 for explanation.

21. Line 282-283: To support the conclusion that the ships are the main source for NO2, the authors should roughly estimate the influence of the surrounding emission sources (especially the main roads and highway, because this could be a significant NO2 source).
R: Due to the lack of relevant research in the areas of Wusong site, we are unable to obtain the influence of the surrounding emission sources accurately. We have installed two active LP-DOAS (Long-Path DOAS) in the Wusong MSB and Fudan University Jiangwan Campus in March of 2018, respectively. The locations of Wusong and Fudan have been shown in Figure R7 (a), and the red arrow indicates the light path of the two LP-DOAS. The campus is 4 kilometers away from Wusong and is considered to be free of pollution because the campus is almost covered by green spaces and have no major emission sources.
Considering the synchronization of data, concentration of NO₂ from March 15 to March 30, 2018 have been analyzed. During the observation, the average value of NO₂ in Fudan campus and Wusong site is 12.42 ppb and 30.50 ppb, respectively. In order to make a clearer explanation, we have shown the time series of NO₂ in a short segments of three days as an example in Figure R7 (b). We have calculated the difference of NO₂ between Wusong and Fudan campus, which is considered to represent the sum of NO₂ emissions from ships and surrounding sources at Wusong area. For the red line in Figure R7 (b), the rapidly changing part is discharged by the ships, while the smooth part may come from surrounding emission sources such as roads and highway. After a rough estimation, the proportion of NO₂ emitted by ships is more than 47%. However, this estimation are quite rough and more accurate conclusion need be furthered with multiple measurements and technical method.
→ Ok, but the authors should make a short note in the manuscript that the reader can follow this explanation

23. Line 292: largest container ports related to what? China, Asia, World?
R: Yantian Port in Shenzhen is the largest single port area with the largest container throughput in China. According to the survey, Shenzhen Port is the third port of the world container port in 2017, and Yantian Port is the main port of Shenzhen Port. So the Yantian Port is one of the largest container port in China and even in the world.
→ Ok, but the authors should also mention in the manuscript that Yantian Port is one of the largest container ports in China and even in the world

Reviewer #2

3. In section 2.3, the SO2 and NO2 DSCDs are retrieved at different spectral ranges. How do the authors compensate the effect of wavelength dependency? If it is not considered in the retrieval, an error analysis is required.
R: The configuration of SO₂ and NO₂ spectral analysis was based on many previous studies, e.g. Hendrick et al., 2014; Irie et al., 2011; Seyler et al., 2017; Wang et al., 2014. So the common fitting window of 307.5-320 nm and 338-370 were used for SO₂ and NO₂, respectively. As it can be seen in Fig. R3, the strong absorption band of SO₂ is below 325 nm, where the NO₂ absorption are relatively weak. It means that the wavelength band of SO₂ analysis window should be shorter than that of NO₂.

[Figure]

*Figure R3. Absorption cross section of NO₂ and SO₂ in the wavelength range of 300~400 nm.*

Since it is obvious that the SO₂ analysis cannot be performed well in longer wavelength over 325 nm, we have tried the analysis of NO₂ with the same fitting interval of SO₂ in 307.5~320 nm. As shown in the Fig. R4 (a), we found that the NO₂ DSCD values from fitting window of 307.5~320 nm are larger than that in 338-370 nm and simultaneously shows considerable uncertainties. In addition, Fig. R4 (b) and (c) show that fitting interval of 307.5~320 nm for NO₂ generates even larger RMS and DSCDs error compared to the results from fitting within 338~370 nm. It suggests that the DSCDs from same fitting window will bring large uncertainty and error in the results. Finally, we decided to use the different fitting intervals for SO₂ and NO₂.

[Figure]

*Figure R4. Comparison of NO₂ retrieval with different fitting intervals of 307.5-320 nm and 338-370 nm on 26 June 2018: (a) NO₂ DSCD with error bars, (b) RMS and (c) DSCD error.*

Regarding to the compensation of wavelength dependency effect, we think the way to use the ratio of SO₂ to NO₂ DSCDs to identify the ship emission will not be impacted by the effect of wavelength dependency. Because the fixed analysis fitting window was applied for all campaigns and the ratio will not contain the wavelength dependency effect (or in presence as the systematic deviations).

→ Ok, but the authors should add at least some of this response to the manuscript to make it more robust.

---

## Author Response (AR2)

**Response to the comments from reviewer and editor**

The initial reviewer's comments are indicated in black, the authors' reply in blue. After the second round of review, the comments from editor and reviewer to the authors are shown in green, and the author's second reply in pink.

**Editor's comments:**
Comments to the Author:

Dear Authors,
Thank you very much again for your comprehensive explanations and answers to the reviewer questions related to the initially submitted version of the manuscript. Reviewer #1 (and the editor) is quite satisfied with the given answers. Also the points of reviewer #2 have been adequately addressed. The manuscript is on a good way, the revised version shows considerable improvement. Nevertheless, some content of your replies did not enter the revised manuscript, but it should to make a few points clearer. So your paper is "accepted subject to minor changes".
As your answers given in your response reports are of good quality you should take up some of the formulated content (in a condensed manner) into the manuscript. You can find good hints for this in the new referee report (09/09/19). There, reviewer makes clear suggestions (colour coded in green) how to update your manuscript. Please follow this suggestion to generate the final manuscript.

Non-public comments to the Author:

Dear authors, please try to follow the suggestions made in the referee report to extend a little your manuscript. That means, take up some of the points formulated in your answers to the referees into the manuscript. The referee gives good hints for this in his report (colour coded in green). Try to be concise.
So everything is there, it just needs to find its way into the manuscript.
R: It's very delighted to receive the comments from editor. Accordingly, we have made related revision of the manuscript by following the referee report. Please refer to the detailed response to the reviewer.

**Referee Report (**09/09/19**):** Referee comments on the revised manuscript "Surveillance of SO2 and NO2 from ship emissions by MAX-DOAS measurements and implication to compliance of fuel sulfur content" of Cheng et al., 2019 for submission in ACP.
The authors addressed all comments and suggestions in an adequate way. The corrections significantly improve the quality of the revised manuscript. However, the authors reply to some of the reviewer's comments in detail only in the response to comments file, but did not update the manuscript. As the explanations in the reply are of good quality which does improve the understanding of the manuscript, I do recommend considering the manuscript for publication in ACP with the below minor changes:
I do reply to selected authors reply on the first reviewer #1 and #2 comments.
R: Thanks again to the reviewer for helpful suggestions, which are very positive on the scientific content of the manuscript and make the content more robust and persuasive. Based on these suggestions, we have added some short notes and explanations to the manuscript. We addressed all the minor revision for consideration as below. All line numbers mentioned below refer to the modified original without tracking changes.

Reviewer #1
3. Results presented in section 3.1 do represent only one measurement at one day. If the authors have "more than a dozen cycles" (line 220-221), why they don't show an average of all cycles. The presented cycle is a snapshot and might be not representative to draw conclusions. Well, even an average over one afternoon does represent only a snapshot of the situation. The authors should consider this in their discussion or should skip this section.
R: There may be some misunderstandings due to the unclear description. The purpose of Section 3.1 is to investigate the 2-D distribution of retrieved $NO_2$ and $SO_2$ DSCDs using 2-D scanning measurement of MAX-DOAS. The hotspots of $NO_2$ and $SO_2$ are related to the azimuth of the berth where the ship is docked and the corresponding ship operation status. It is obviously that the 2-D distribution of $NO_2$ and $SO_2$ DSCDs changes with time, so that it seems unreasonable to show the average of all scan cycles, which may weaken the spatial distribution of hotspots. However, the results of the rest of the cycles were also shown in Figure 6. It can be seen that the concentration at a given azimuth and elevation is constantly changing throughout the day.
→ The authors could summarize this description and add it to the manuscript introduction.
$2^{nd}$ R: Considering the hotspots of $NO_2$ and $SO_2$ in each 2-D distributions are related to the azimuth of the berth where the ship is docked and the corresponding ship operation status, it is not suitable to present in averaged values. The results of each cycles during the afternoon were shown in Fig. 6.
We have summarized this description and added it to the section 3.1. Please refer to Line 236-239, i.e. "In view of the identified emission source position above, considering that the hotspots of NO2 and SO2 in each 2-D distributions are related to the azimuth of the berth where the ship is docked and the corresponding ship operation status, the DSCDs of NO2 and SO2 observed at elevation 4° and azimuth angle between 31°-33° were selected to display the temporal pattern of emissions at berth without averaging."

11. Line 190: What is located opposite the berth? Any industry which might emit NOx or SO2?
R: Opposite the berth is the South Channel of the Yangtze Estuary. The only sources of NOx and $SO_2$ could be the ships emissions on the channel, and the main channel is more than two kilometers away from the berth. Behind the building of Pudong MSB, there are green land and residential area. The container yard was located between the building and berths.

→ Ok, but the authors should add a short statement to the manuscript that there are no other sources for SO2 and NO2 than emissions from ships.

2nd R: We have improved it and added the short statement to the manuscript. Please refer to Line 203-204, i.e. "There are no other obvious sources for SO2 and NO2 than emissions from ships."

16. Line 253: ships in navigation or maneuver? Do the authors can exclude emissions from the industry behind the river (visible in Fig 2 (c)) as a source?

R: The ships were in the state of navigation. According to Figure R4, we can see the main green land areas behind the river, including villages and forest parks. In addition, the opposite of the river is a small dock and a station for transporting containers. Due to the lack of relevant research and observations in this region, it is very difficult to estimate the amount of NO2 and SO2 emitted from the opposite side, and we have to ignore this part of the source compared with the ship emissions on the channel.

→ Ok, but if the authors cannot exclude this influence the measurements, they have to make a short note in the manuscript. E.g. "A small dock and a station for transport container are located opposite the river, which might slightly influence the measurements." or something similar.

2nd R: We have improved it. The short note have been added. Please refer to Line 270-271, i.e. "The observed signal of pollutants mainly come from the emissions of ships in navigation. Nevertheless, a small dock and a station for transport container are located opposite the river, which might slightly influence the measurements."

**Figure R6. Relationship between DSCDs of (a) NO2 and (b) SO2 and ship density.**

Corresponding to Fig. 9 in the revised manuscript

→ In the figure caption the authors have to explain the meaning of the squares, stars and bars or have to refer to Figure 7 for explanation.

2nd R: We have added the explanations in the figure caption of Fig. 9. Please refer to Line 320-323.

21. Line 282-283: To support the conclusion that the ships are the main source for NO2, the authors should roughly estimate the influence of the surrounding emission sources (especially the main roads and highway, because this could be a significant NO2 source).

R: Due to the lack of relevant research in the areas of Wusong site, we are unable to obtain the influence of the surrounding emission sources accurately. We have installed two active LP-DOAS (Long-Path DOAS) in the Wusong MSB and Fudan University Jiangwan Campus in March of 2018, respectively. The locations of Wusong and Fudan have been shown in Figure R7 (a), and the red arrow indicates the light path of the two LP-DOAS. The campus is 4 kilometers away from Wusong and is considered to be free of pollution because the campus is almost covered by green spaces and have no major emission sources. Considering the synchronization of data, concentration of NO2 from March 15 to March 30, 2018 have been analyzed. During the observation, the average value of NO2 in Fudan campus and Wusong site is 12.42 ppb and 30.50 ppb, respectively.

In order to make a clearer explanation, we have shown the time series of $NO_2$ in a short segments of three days as an example in Figure R7 (b). We have calculated the difference of $NO_2$ between Wusong and Fudan campus, which is considered to represent the sum of $NO_2$ emissions from ships and surrounding sources at Wusong area. For the red line in Figure R7 (b), the rapidly changing part is discharged by the ships, while the smooth part may come from surrounding emission sources such as roads and highway. After a rough estimation, the proportion of $NO_2$ emitted by ships is more than 47%. However, this estimation are quite rough and more accurate conclusion need be furthered with multiple measurements and technical method.

→   Ok, but the authors should make a short note in the manuscript that the reader can follow this explanation.

$2^{nd}$ R: We have made the short note in the manuscript that the amount of $NO_2$ contributed by these sources cannot be ignored. Please refer to Line 332-333, i.e. "It is another shortcoming of this measurement that the MAX-DOAS measured NO2 are considerably impacted by surrounding other emission sources, such as main roads and highways nearby, which contribute un-ignored amounts to the ambient NO2."

23. Line 292: largest container ports related to what? China, Asia, World?

R: Yantian Port in Shenzhen is the largest single port area with the largest container throughput in China. According to the survey, Shenzhen Port is the third port of the world container port in 2017, and Yantian Port is the main port of Shenzhen Port. So the Yantian Port is one of the largest container port in China and even in the world.

→   Ok, but the authors should also mention in the manuscript that Yantian Port is one of the largest container ports in China and even in the world.

$2^{nd}$ R: We have mentioned in the manuscript that Yantian Port is one of the largest container ports in China and even in the world. Please refer to Line 343.

Reviewer #2

3. In section 2.3, the SO2 and NO2 DSCDs are retrieved at different spectral ranges. How do the authors compensate the effect of wavelength dependency? If it is not considered in the retrieval, an error analysis is required.

R: The configuration of $SO_2$ and $NO_2$ spectral analysis was based on many previous studies, e.g. Hendrick et al., 2014; Irie et al., 2011; Seyler et al., 2017; Wang et al., 2014. So the common fitting window of 307.5-320 nm and 338-370 were used for $SO_2$ and $NO_2$, respectively. As it can be seen in Fig. R3, the strong absorption band of $SO_2$ is below 325 nm, where the $NO_2$ absorption are relatively weak. It means that the wavelength band of $SO_2$ analysis window should be shorter than that of $NO_2$.

[Figure]

***Figure R3. Absorption cross section of NO₂ and SO₂ in the wavelength range of 300~400 nm.***
Since it is obvious that the SO₂ analysis cannot be performed well in longer wavelength over 325 nm, we have tried the analysis of NO₂ with the same fitting interval of SO₂ in 307.5~320 nm. As shown in the Fig. R4 (a), we found that the NO₂ DSCD values from fitting window of 307.5~320 nm are larger than that in 338-370 nm and simultaneously shows considerable uncertainties. In addition, Fig. R4 (b) and (c) show that fitting interval of 307.5~320 nm for NO₂ generates even larger RMS and DSCDs error compared to the results from fitting within 338~370 nm. It suggests that the DSCDs from same fitting window will bring large uncertainty and error in the results. Finally, we decided to use the different fitting intervals for SO₂ and NO₂.

[Figure]

***Figure R4. Comparison of NO₂ retrieval with different fitting intervals of 307.5-320 nm and 338-370 nm on 26 June 2018: (a) NO₂ DSCD with error bars, (b) RMS and (c) DSCD error.***
Regarding to the compensation of wavelength dependency effect, we think the way to use the ratio of SO₂ to NO₂ DSCDs to identify the ship emission will not be impacted by the effect of wavelength dependency. Because the fixed analysis fitting window was applied for all campaigns and the ratio will not contain the wavelength dependency effect (or in presence as the systematic deviations).

→   Ok, but the authors should add at least some of this response to the manuscript to make it more robust.

2nd R: Thanks for the comments. We have improved it and added a short explanation to the Sect. 2.2 in the manuscript. Please refer to Line 154-155 and 156-158.

In addition, we have made several technical corrections to the manuscript.

1. There is a clerical error in the Line 156, the "405-430 nm" has been changed to "338-370 nm".

2. Line 304, in order to keep in a uniform format, the "1.5" has been changed to "1.50".

3. Line 627, the name of journal "Atmospheric Chemistry and Physics" was repeated and has been deleted.

[revised manuscript text omitted]